

# Soil methane oxidation in both dry and wet temperate eucalypt forests show near identical relationship with soil air-filled porosity

Benedikt J. Fest[1], Nina Hinko-Najera[2], Tim Wardlaw[3], David W.T. Griffith[4], Stephen J. Livesley[1], Stefan K. Arndt[1]

[1]School of Ecosystem and Forest Sciences, The University of Melbourne, Richmond, 3121 Victoria, Australia
[2]School of Ecosystem and Forest Sciences, The University of Melbourne, Creswick, 3363 Victoria, Australia
[3]Forest Management Services Branch, Forestry Tasmania, Hobart, 7000 Tasmania, Australia
[4]School of Chemistry, University of Wollongong, Wollongong, 2522 New South Wales, Australia

*Correspondence to*: Benedikt J. Fest (bfest@unimelb.edu.au)

**Abstract.** Well-drained, aerated soils are important sinks for atmospheric methane ($CH_4$) via the process of $CH_4$ oxidation by methane oxidising bacteria (MOB). This terrestrial $CH_4$ sink may contribute towards climate change mitigation, but the impact of changing soil moisture and temperature regimes on $CH_4$ uptake is not well understood in all ecosystems. Temperate eucalypt forests in south-eastern Australia are predicted to experience rapid and extreme changes in rainfall patterns, temperatures and wild fires. To investigate the influence of environmental drivers on seasonal and inter-annual

variation of soil-atmosphere $CH_4$ exchange we measured soil-atmosphere $CH_4$ exchange at high temporal resolution (<2 hr) in a dry temperate eucalypt forest in Victoria (Wombat State Forest, 870 mm $yr^{-1}$) and in a wet temperature eucalypt forest in Tasmania (Warra LTER, 1700 mm $yr^{-1}$). Both forest soil systems were continuous $CH_4$ sinks of -1.79 kg $CH_4$ $ha^{-1}$ $yr^{-1}$ in Victoria and -3.83 kg $CH_4$ $ha^{-1}$ $yr^{-1}$ in Tasmania. Soil $CH_4$ uptake showed substantial temporal variation and was strongly controlled by soil moisture at both forest sites. Soil $CH_4$ uptake increased when soil moisture decreased, and this relationship

explained up to 90% of the temporal variability. Furthermore, the relationship between soil moisture and soil $CH_4$ flux was near identical at both forest sites when soil moisture was expressed as soil air-filled porosity (AFP). Soil temperature only had a minor influence on soil $CH_4$ uptake. Soil nitrogen concentrations were generally low, and fluctuations in nitrogen availability did not influence soil $CH_4$ uptake at either forest site. Our data indicate that soil MOB activity in the two forests was similar and that differences in soil $CH_4$ exchange between the two forests were related to physiochemical differences in

soil properties influencing soil gas diffusivity. The differences between forest sites and the variation in soil $CH_4$ exchange over time could be explained by soil air-filled porosity as an indicator of soil moisture status.



## 1 Introduction

Methane ($CH_4$) has a relatively low atmospheric concentration of approximately 1.8 ppm and is after carbon dioxide ($CO_2$, approx. 402 ppm) the second most abundant greenhouse gas in the atmosphere (IPCC, 2013). Although its atmospheric concentration is two orders of magnitude lower than that of $CO_2$, $CH_4$ accounts for approximately 18% of the currently
observed global temperature increase (IPCC, 2013). In addition, $CH_4$ contributes to 32% of the current radiative forcing created by the major greenhouse gases as it has a 25 times greater global warming potential (GWP) compared to $CO_2$ (IPCC, 2013).

Forest soils are the most important land based sink for $CH_4$ via the activity of methane oxidising bacteria (MOB) in well-drained, aerobic soils. Soils in temperate forest ecosystems play an important role in global $CH_4$ exchange between the land
mass and the atmosphere, and they constitute around 30-50% of the soil based $CH_4$ sink worldwide (Ojima et al., 1993;Dutaur and Verchot, 2007).

Major environmental factors controlling and influencing $CH_4$ uptake rates by forest soils are soil diffusivity and -structure, soil moisture, soil temperature and soil nitrogen status (Ball et al., 1997a;Smith et al., 2003;von Fischer and Hedin, 2007;Butterbach-Bahl et al., 2002;Del Grosso et al., 2000).
The main factor regulating the $CH_4$ uptake capacity of soils is the diffusion rate of $CH_4$ through the soil and hence the substrate availability of $CH_4$ to the MOB across the soil profile. $CH_4$ uptake rates have been shown to decrease with increasing soil moisture as a result of decreasing soil gas diffusion rates across different ecosystems (Castro et al., 1995;Khalil and Baggs, 2005;Ball et al., 1997). Therefore, $CH_4$ uptake is thought to be most rapid in coarse-textured forest soils with a well-developed structure and an organic surface layer that does not inhibit gas diffusion (Boeckx et al., 1997;Del
Grosso et al., 2000;Smith et al., 2000). Soil bulk density can also correlate with soil $CH_4$ uptake across different ecosystems (Smith et al., 2003;Smith et al., 2000), which is not unexpected since soil air filled porosity, which is directly linked to soil diffusivity, is a function of soil bulk density and volumetric water content.

Soil $CH_4$ uptake at atmospheric levels generally shows limited temperature dependency and reported $Q_{10}$ values are generally low with an average around 1.4 (Crill, 1991;Born et al., 1990;Smith et al., 2000). Another factor that influences the
$CH_4$ uptake capacity of soils is soil N status, especially the availability of ammonium ($NH_4^+$) (Butterbach-Bahl et al., 1998;Sitaula et al., 1995). Increasing soil N availability through organic and inorganic fertiliser additions and through biological N fixation can decrease $CH_4$ uptake rates (Niklaus et al., 2006;Dick et al., 2006).

Temperate eucalypt (broadleaved evergreen) forests in south-eastern Australia cover around 26 million hectares (Committee, 2013), and provide a large range of ecosystem services. However, despite a growing interest in soil $CH_4$ uptake in the last
decade there have been very few studies investigating $CH_4$ oxidation in soils of natural Australian forest and woodland ecosystems with only a relatively small number of published studies on $CH_4$ uptake in temperate forest systems (Fest et al., 2009;Livesley et al., 2009b;Meyer et al., 1997;Fest, 2013;Fest et al., 2015b;Fest et al., 2015a), tropical forest systems (Kiese et al., 2003) and savanna ecosystems (Livesley et al., 2011). Moreover, there is currently no model that accurately predicts



the size of the terrestrial $CH_4$ sink in Australia or determines how the strength of this sink will change over time. Data describing $CH_4$ emission and oxidation from Australian soils is still patchy and often lacking for important landscapes such as tropical savannas, the semi-arid and arid zones and woody ecosystems (Dalal et al., 2008).

Compared to most European and North American temperate forest systems, forest soils in the Australian temperate region
are generally highly weathered and very low in nutrients, and atmospheric nitrogen deposition is very low. Furthermore most of the temperate forest area in Australia does not experience periods of snow cover or below zero soil temperatures. It is therefore questionable as to whether information gathered on spatial and temporal variability of soil $CH_4$ exchange in Northern Hemisphere temperate forest soils are transferable to those in Australia. Furthermore, it is not clear if processes that explain soil $CH_4$ uptake in deciduous forest systems or coniferous forest systems worldwide can be directly transferred to the
eucalypt or acacia forest systems that dominate the forests and woodlands of Australia. Most estimates of soil $CH_4$ exchange in Australian forest systems were based on infrequent (weekly-monthly) or campaign-based measurements (of one to two weeks), which may not fully reflect the temporal dynamics and range of environmental conditions.

This study investigates soil-atmosphere $CH_4$ exchange using automated chamber systems measuring at a high temporal resolution over 1-2 years in two temperate *Eucalyptus obliqua* dominated forests sites with contrasting annual precipitation.
The main objectives of this study were to assess the magnitude and temporal variation in $CH_4$ exchange between the soil and atmosphere in temperate evergreen eucalypt forest systems and to investigate the primary biophysical processes that control the seasonality in soil $CH_4$ flux.

## 2 Material and Methods

### 2.1 Site description

The Tasmanian site is in the Warra Long Term Ecological Research (LTER) Site approximately 60 km west-southwest of Hobart, Tasmania, Australia (**AU-Wrr**: 43° 5'36.78''S, 146° 38'42.65''E). The site is dominated by *Eucalyptus obliqua* (L'Herit.) with an overstorey height of around 53 m and a basal area of 54 $m^2$ $ha^{-1}$. The understorey is mainly comprised of *Acacia melanoxylon* (R.Br.), *Nothofagus cunninghamii* (Hook.) Oerst. and *Dicksonia antarctica* (Labill.). The climate of AU-Wrr is classified as temperate cool wet (Dunlop and Brown, 2008) with cold and wet winters and warm and wet
summers. The average rainfall is approximately 1700 mm $yr^{-1}$ (Fig. 1a) with mean monthly maximum temperatures of 19.3 °C in January (summer) and mean minimum temperatures of 2.5 °C in July (winter). The soils at Warra are derived from Permian siltstone with a surface texture of silty loam to silty clay loam, and are classifed as kurosolic redoxic hydrosol (McIntosh, 2012). The average bulk denisty in the top 5 cm of mineral soil is 0.67 g $cm^{-3}$ and soil porosity is 0.74 $cm^3$ $cm^{-3}$.

The Victorian forest site is in the Wombat State Forest, approximately 120 km west of Melbourne, Australia (**AU-Wom**: 37°
25'20.83''S, 144° 5'38.63''E). AU-Wom is dominated by *Eucalyptus obliqua* (L. Her.)*, Eucalyptus rubida* (H. Deane & Maiden) and *Eucalyptus radiata* (Sieber ex DC) trees of approximately 20 – 25 m in height and 37 $m^2$ $ha^{-1}$ of stem basal area. The climate is classified as Mediterranean to cool temperate, with warm and dry summers and wet and cool winters.



The average rainfall is approximately 870 mm yr$^{-1}$ Fig. 1b,) with mean monthly maximum temperatures of 25.6 $^{o}$C in January (summer) and mean minimum temperatures of 3.4 $^{o}$C in July (winter). The soils of AU-Wom are derived from weathered sandstone and shale, with a surface texture of sandy clay loam, classified as an acidic-mottled, dystrophic, yellow Dermosol (Robinson et al., 2003). The average bulk density in the top 5 cm of mineral soil is 0.90 g cm$^{-3}$ and soil porosity is 5   0.65 cm$^{3}$ cm$^{-3}$.

### 2.1.1 Experimental design AU-Wrr

The temporal variation in soil-atmosphere exchange of $CH_4$ was monitored continuously from 10/10/2010 to 15/01/2012 using a fully-automated gas chromatograph (GC) measurement system attached to ten pneumatic open-and-close chambers as described in Livesley et al. (2009). This system was supported by a remote area power system consisting of a 5kV diesel 10   generator and 12V battery bank. The ten chambers were randomly distributed over an area of approximately 25 x 25 m. Chambers were attached to a square steel frame base (e.g. 50 cm x 50 cm) which was inserted 5 cm into the soil, and a plexiglass headspace of 15 cm depth (e.g. 37.5 L chamber volume). Chambers were attached to the frame using clamps and closed cell foam. For each chamber, six flux rate measurements were made during a 24 hour period, one every four hours. Further details of the automated trace gas measurement system, chamber design and gas chromatograph can be found in 15   Butterbach-Bahl et al. (1997);Papen and Butterbach-Bahl (1999) and Livesley et al. (2009). Soil temperature (12-Bit Temp Smart Sensor, Onset Computer Cooperation, USA) and moisture (EC-5 Soil Moisture Smart Sensor, Onset Computer Cooperation, USA) was logged at 0-10 cm on a half hourly basis (Hobo U30, Hobo Data Logger, Onset Computer Cooperation, USA) in the middle of the site. Chamber pneumatic lids opened automatically when rainfall, measured by a tipping bucket rain gauge, exceeded 1 mm in 5 minutes to avoid a potential reduction in soil moisture inside the chambers 20   caused by the rainfall exclusion during the relatively long time of chamber closure (2h).

### 2.1.2 Experimental design AU-Wom

Temporal variation in soil-atmosphere exchange of $CH_4$ was monitored continuously from 1/5/2010 to 30/04/2012 using a fully-automated Fourier Transform Infrared (FTIR) spectrometer measurement system attached to six pneumatic open-and-close chambers (Griffith et al 2012). This system was supported by a remote area power system consisting of a 4.5kV diesel 25   generator and 24V battery bank. The automatic chambers used followed the same design as that described at the AU-Wrr site. The opening and closing of the lids via pneumatic pistons was controlled with the measuring software on site (PC). Six chambers were distributed randomly over an area of around 25 x 25 m and were measured in sequence with each chamber initially having a measuring period of 15 minutes (1/5/2010 – 21/10/2010) that was later extended to 20 minutes to increase detection precision for other simultaneously measured trace gases (22/10/2010 – 30/04/2012). Lids were open for the first 2 30   and the last 2 minutes of every 15/20 minute measuring interval per chamber to flush the sample lines with ambient air resulting in a chamber incubation period of 11/16 minutes. One $CH_4$ flux measurement per chamber was achieved every 1.5/2 hours. The chambers were not fitted with a fan, but there was forced ventilation during the incubation period of each



chamber through the use of an external pump which circulated the air in a closed loop through the head-space of the chamber (closed dynamic setup), attached airlines (0.3 L tubing volume) and the measuring cell (3.5 L cell volume) of a Fourier Transform Infrared (FTIR) spectrometer setup (Spectronus, ECOTECH P\L, Australia). The spectrometer (Bruker IRcube with globar source and thermoelectrically cooled MCT detector) measured concentrations of $CH_4$, $CO_2$, $N_2O$, carbon

monoxide and water vapour in the air stream (Meyer et al., 2001;Griffith et al., 2012;Hammer et al., 2012). Measurements of the $CH_4$ concentration were made every minute during the 15/20 min chamber period. Further information about measuring principle, instrument setup, maintenance and calibration can be found in Griffith et al. (2012). Soil temperature (Thermocouple Probe) and moisture (impedance probes, ML2x – Theta Probe Soil Moisture Sensor, Delta-T Devices LTD, UK) was recorded continuously at 0-5 cm within each chamber. In addition, soil temperature (Averaging Soil Thermocouple

Probe, TCAV, Campbell Scientific, Australia, Pty Ltd) and soil moisture (Water Content Reflectometer, CS616, Campbell Scientific, Australia, Pty Ltd) were recorded on a half hourly basis at 0-10 cm by an onsite eddy covariance system. Given the relatively short closure period of 11/16 minutes for each chamber during a 4 hour period, we decided that automated chamber opening in response to rainfall events was not necessary.

**2.2 Flux calculation**

$CH_4$ flux rates were calculated for both automated measuring systems from the rate of increase/decrease of gas concentration in the chamber head space with time according to:

$$F_{\mu L} = (V/A) \times (dC_{CH4}/dt) \tag{1}$$

Where $V$ is the volume (L) of the chamber head space plus sample lines and the FTIR sample cell, $A$ is the soil surface area covered by the chamber ($m^2$) and $t$ is time. The term $dC_{CH4}/dt$ ($\mu L\ L^{-1}\ h^{-1}$) was calculated from the initial linear $CH_4$ concentration change after chamber closure. In cases where the fitted linear regression model had an $R^2 < 0.9$ then this flux measurement was excluded from further analysis. The determined flux rate ($F_{\mu L}$) was subsequently converted to $\mu mol\ CH_4$ $m^{-2}\ h^{-1}$ ($F_{\mu mol}$) by accounting for temperature, pressure and volume using Equation (2) based on the ideal gas law:

$$F_{\mu mol} = (F_{\mu L} \times P) / (R \times T) \tag{2}$$

Where $P$ is the atmospheric pressure in kPa at site according to altitude or direct measurement (Eddy tower), $R$ is 8.3144 (the ideal gas constant in $L\ kPa^{-1}\ K^{-1}$), and $T$ is the air temperature in Kelvin (273.15 + $^oC$). Fluxes in $\mu mol\ CH_4\ m^{-2}\ h^{-1}$ were then

converted to $\mu g\ CH_4$-$C\ m^{-2}\ h^{-1}$ based on the molecular atomic mass.



## 2.3 Additional measurements

From within each site, composite soil samples (three 0-5 cm samples) were collected, sieved (2 mm) and sub-sampled for 1M KCl extraction (1:4, soil:KCl) and gravimetric water content ($GWC_S$) determination (105°C for 48 hours) during additional seasonal measurement campaigns spread across the measurement timeframe (n = 13 in AU-Wom, n = 10 in AU-

Wrr). KCl extracts were filtered (Whatman 42) and frozen prior to analysis for nitrate ($NO_3^-$) and ammonium ($NH_4^+$) concentration using an auto-analyser (SFA, Technicon ™).

During initial site installation (and over the course of the measurement timeframe) approximately 30 volumetric soil cores (0-5 cm, Ø 72 mm) were sampled at each site to determine soil volumetric water content (VWC) and soil bulk density (BD). The data were used to establish site dependent calibration curves between the onsite installed soil moisture sensors (HOBO

Micro Station Data logger H21 and EC-5 Soil Moisture Smart Sensor, Onset Computer Corporation, USA), hand held impedance probes (ML2× Theta probe and HH2 Moisture Meter, Delta-T Devices Ltd, UK) and VWC (Kaleita et al., 2005). The bulk density and volumetric water content data and their relationship to the onsite installed soil moisture sensor readings and hand held impedance probes readings were further used to calculate soil porosity, air filled porosity and percentage water filled pore space (%WFPS) for each plot and measuring event according to Loveday and Commonwealth Bureau of

Soils (1973):

$$\text{Soil porosity} = 1 - (\text{soil bulk density} / \text{particle density}) \tag{3}$$

Where a value of 2.65 was used for particle density (g cm$^{-3}$) of rock, sand grains and other soil mineral particles.

$$\text{Air filled porosity} = \text{Soil porosity} - \text{volumetric water content} \tag{4}$$

$$\%\text{WFPS} = (\text{volumetric water content} \times 100) / \text{Soil porosity} \tag{5}$$

At the end of the study, a composite soil sample from five soil cores was collected at 0-5 cm at each site, air dried, sieved (2 mm) and analysed for soil particle size analysis through dispersion, suspension, settling and sequential hydrometer readings (Ashworth et al., 2001). A sub-sample of each air-dried soil was analysed for pH (1:5, soil:water) and for total C and N content using an elemental analyser (LECO®).

## 2.4 Data presentation and statistical analyses

Flux and environmental sensor data presented (if not specifically related to individual chambers) in the figures here after are averages for respective chamber cycles where at least 2/3 of the chamber flux measurements had passed the above mentioned flux quality control (1.5/2 hour cycle for the FTIR system and a 4 hour cycle average for the GC system) at each



site ±1 SE (where error bars are present). We also calculated the coefficient of variance per chamber cycle (CV%$_{cycle}$) by dividing the standard deviation of each chamber cycle by its respective mean and multiplying the result by 100. Furthermore, soil temperature and soil moisture data were averaged accordingly for each chamber cycle to allow regression analysis. In a second step, to enable correlation analysis with daily rainfall and sporadic soil inorganic nitrogen measurements we

calculated daily site averages of the measured fluxes and environmental parameters, with the exception of rainfall where we calculated daily sums, for days where at least 80% of chamber cycles were available. We additionally calculated the coefficient of variation per day (CV%$_{day}$) for the CH$_4$ flux data.

All statistical analyses were performed with SPSS 20 (IBM, USA). Linear regression procedures and multiple linear regression procedures were used to investigate temporal relationships between measured soil environmental parameters and

soil CH$_4$. We initially ran stepwise linear regression procedure as an exploratory tool to identify significant predictors and predictor combinations and retested these afterwards in simple or multiple linear regression models. We transformed data when necessary to reduce heteroscedasticity for linear regression analysis.

### 2.5 Annual site CH$_4$ flux budgets

To calculate annual site CH$_4$ flux budgets for both sites we first selected a 12 month period with the greatest data coverage

for daily average flux for both sites (1/1/2011 – 1/1/2012) and filled existing flux data gaps as follows. For small data gaps of single days where no environmental sensor or flux data were available, we calculated values based on linear interpolation between the CH$_4$ flux of the day before the gap and the day after the gap. For data gaps longer than one day, we used the linear regression model between soil VWC soil moisture and daily soil CH$_4$ flux for each site (Table 1) to estimate the missing CH$_4$ flux data.

## 3 Results

### 3.1 CH$_4$ flux in relation to soil environmental variables

At the AU-Wrr site, soil CH$_4$ flux was always negative indicating CH$_4$ uptake all year around (Fig. 2). The measurement cycle means ranged between -2 µg CH$_4$ m$^{-2}$ h$^{-1}$ (spring 2010) to -58.4 µg CH$_4$ m$^{-2}$ h$^{-1}$ (autumn 2011) with an arithmetic mean of -41.2 ± 0.23 SE µg CH$_4$ m$^{-2}$ h$^{-1}$. In general, months with higher average soil moisture and higher total rainfall displayed

lower CH$_4$ uptake when compared to months with lower average soil moisture and lower total rainfall (Fig. 2). Inter-annual differences in average site CH$_4$ uptake between seasons (spring and summer) were also reflected in concurrently recorded average site soil moisture levels. The coefficient of variance (CV) for the average CH$_4$ flux based on 10 chambers in one measurement cycle ranged between 1.8 and 98.0% with an average of 17.9 ± 0.23% (SE) and was higher in periods of rapid changes in soil moisture levels reflecting changes in precipitation (Fig. 2). The linear regression analysis showed that

volumetric water content (VWC) accounted for around 85% of variability in soil CH$_4$ uptake across all seasons (Fig.4a, Table 1) with soil CH$_4$ uptake decreasing when soil VWC increased or soil CH$_4$ uptake increasing when air filled porosity





(AFP) increased (Fig.4b, Table 1). Soil temperature (0-5 cm) alone was weakly related to $CH_4$ uptake with higher $CH_4$ uptake rates associated with higher soil temperatures. However, soil temperature alone was only able to account for around 19% of seasonal variability in $CH_4$ uptake (Fig.4c, Table 1). In addition, after taking the effect of VWC into account, soil temperature only explained around 1.5% of the remaining variability in $CH_4$ uptake at AU-Wrr (data not shown). A regression model containing VWC and soil temperature as input variables had only a marginally higher coefficient of determination when compared to the model only containing VWC (Table 1). AFP or VWC showed some weak dependency of soil temperature at the site ($R^2 = 0.14$, $p < 0.001$).

At the AU-WOM site soil $CH_4$ flux was also always negative indicating $CH_4$ uptake all year around (Fig. 3). The measurement cycle means ranged between -1.3 µg $CH_4$ $m^{-2}$ $h^{-1}$ (recorded during a period of heavy rainfall in summer 2011) to -62.5 µg $CH_4$ $m^{-2}$ $h^{-1}$ (summer 2010) with an arithmetic mean of -25.5 ± 0.16 SE µg $CH_4$ $m^{-2}$ $h^{-1}$. Similar to the AU-WRR site months with higher average soil moisture and higher total rainfall displayed lower $CH_4$ uptake when compared to months with lower average soil moisture and lower total rainfall (Fig. 3). The CV for the average $CH_4$ flux based on 6 chambers in one measurement cycle ranged between 6.7 and 143.0% with an average of 29.3 ± 0.12% (SE) and was again higher in times of rapid soil moisture changes in response to changes in precipitation patterns (Fig. 3). Furthermore, inter-annual differences in average site $CH_4$ uptake between seasons were also reflected in concurrently recorded average site soil moisture levels. The linear regression analysis showed that VWC could account for around 91% of variability in soil $CH_4$ uptake across all seasons (Fig.4a, Table 1) with soil $CH_4$ uptake decreasing when soil VWC increased, the opposite trend was observed for AFP (Fig.4b, Table 1). Soil temperature (0-5 cm) alone was again weakly related to $CH_4$ uptake with higher $CH_4$ uptake rates associated with higher soil temperatures (Fig. 4c). At the AU-WOM site, only around 20% of seasonal variability in $CH_4$ uptake (Table 1) was explained by soil temperature. In addition, similar to the results at AU-Wrr, after taking the effect of VWC into account, soil temperature only explained around 5% of the remaining variability in $CH_4$ uptake at AU-Wom (data not shown). Furthermore, a regression model containing VWC and soil temperature had a marginally lower coefficient of determination (Table 1) when compared to the model only containing VWC (Table 1). AFP or VWC showed some weak dependency of soil temperature at the site ($R^2 = 0.38$, $p < 0.001$)

## 3.2 Mean daily and annual $CH_4$ flux in relation to environmental variables

### 3.2.1 Site AU-Wrr

Daily site averages ranged between -0.12 mg $CH_4$ $m^{-2}$ $d^{-1}$ and -1.35 mg $CH_4$ $m^{-2}$ $d^{-1}$ with an arithmetic mean of -0.98 ± 0.02 SE mg $CH_4$ $m^{-2}$ $d^{-1}$. The coefficient of determination for the regression analysis changed slightly when the regression analysis were calculated on daily means and VWC was able to account for up to 89% in the observed variability in $CH_4$ flux (Table 2). The CV for the daily average site $CH_4$ flux ranged between 0.15% and 20.6% with an average of 3.5 ± 0.19% (SE) and was higher in periods of rapid changes in soil moisture levels. We calculated soil $CH_4$ flux averages for 3 days around the dates when soil $NH_4^+$ and soil $NO_3^-$ samples were taken on-site to enable regression analysis; however neither $NH_4^+$ nor




NO$_3^-$ alone or together could explain any variability in soil CH$_4$ flux at the site and all relationships were non-significant (Fig. 5b, d, f).

### 3.2.2 Site AU-Wom

Daily site averages ranged between -0.11 mg CH$_4$ m$^{-2}$ d$^{-1}$ and -1.36 mg CH$_4$ m$^{-2}$ d$^{-1}$ with an arithmetic mean of -0.62 ± 0.01

SE mg CH$_4$ m$^{-2}$ d$^{-1}$. The CV for the daily average site CH$_4$ flux ranged between 0.11% and 47.6% with an average of 5.6 ± 0.17% (SE) and was again higher in periods of rapid changes in soil moisture levels. As for the AU-Wrr site the coefficient of determination for the regression analysis changed slightly when the regression analysis was calculated on daily means and VWC was able to account for up to 92% in the observed variability in CH$_4$ flux (Table 2). Similar to the AU-Wrr site, three day CH$_4$ flux averages were not significantly correlated with soil NH$_4^+$ or NO$_3^-$ if entered alone or together as predictors to

the linear regression model (Fig. 5a, c, e).

### 3.3 Annual site CH$_4$ flux budgets

The calculated annual CH$_4$ budget for the year 2011 of the AU-Wrr site was -3.83 kg CH$_4$ ha$^{-1}$ yr$^{-1}$. The calculated annual CH$_4$ budget for the year 2011 of the AU-Wom site was -1.79 kg CH$_4$ ha$^{-1}$ yr$^{-1}$.

### 4 Discussion

This study reports continuous measurement of soil-atmosphere CH$_4$ exchange in two temperate eucalypt forests in Australia measured at high temporal resolution for 1-2 years. Mean daily CH$_4$ flux values (AU-Wrr = - 1.35 to -0.12 mg CH$_4$ m$^{-2}$ d$^{-1}$; AU-Wom = - 1.36 to -0.11 mg CH$_4$ m$^{-2}$ d$^{-1}$) were well within the reported range for other temperate forests in Europe (-2.47 to + 0.26 mg CH$_4$ m$^{-2}$ d$^{-1}$; (Smith et al., 2000)) or worldwide (-10.68 to 0.02 mg CH$_4$ m$^{-2}$ d$^{-1}$; (Dalal et al., 2008;Dalal and Allen, 2008)).

The estimated annual CH$_4$ uptake of -1.79 kg CH$_4$ ha$^{-1}$ yr$^{-1}$ for AU-Wom and -3.83 kg CH$_4$ ha$^{-1}$ yr$^{-1}$ for AU-Wrr are comparable to the range of -2.5 to - 3.7 kg CH$_4$ ha$^{-1}$ yr$^{-1}$ reported for temperate beech and spruce forest sites in Germany where CH$_4$ fluxes were measured with a similar automated system over multiple years (Butterbach-Bahl and Papen, 2002). Globally, a range of -1.31 to -10.5 kg CH$_4$ ha$^{-1}$ yr$^{-1}$ has been reported for temperate forest systems based on short and long-term, automated and manual chamber measurement campaigns (Dalal et al., 2008;Dalal and Allen, 2008). The annual CH$_4$

uptake rate estimated for AU-Wom in our study was less than a third of the -5.8 kg CH$_4$ ha$^{-1}$ yr$^{-1}$ estimated by Meyer et al. (1997) for soils in the same forest system. This earlier CH$_4$ sink estimate was based on only five seasonal flux measurements but might also be attributed to the measurements being taken during three dry years (1993 – 1995) when average rainfall was 677 mm yr$^{-1}$ (Meyer et al. (1997). In comparison, the years when our study was undertaken (2010 – 2012) the average rainfall was 1063 mm yr$^{-1}$. This may partly explain the greater CH$_4$ uptake estimate of Meyer et al. (1997)  as the lower soil

moisture levels may well lead to greater CH$_4$ uptake rates.



Given that cool, wet temperate eucalypt forests are often compared to rainforests it is worth noting that the mean annual $CH_4$ uptake estimated for AU-Wrr was similar to that estimated for a tropical rainforest in Queensland, Australia (-3.2 kg $CH_4$ ha$^{-1}$ yr$^{-1}$; Kiese et al., 2003). The soils at AU-Wrr and the Queensland rainforest also have very similar soil characteristics with regards to pH , bulk density and sand, silt and clay fractions (Kiese et al., 2003;Kiese et al., 2008). However, the mean

annual precipitation at the Queensland rainforest site was 2.5 fold greater (4395 mm) than rainfall at the AU-Wrr site, which given similar assumed soil drainage properties indicates large geographical differences in the activity, size and/or structure of the MOB on a continental scale.

One of the most novel results of our study is the strong linear relationship observed between soil moisture and $CH_4$ uptake. To our knowledge the strength of this relationship is unique for temperate forest systems measured using continuous

automated chamber systems over a long-period. It is also striking that this strong linear relationship was similar in the two temperate eucalypt forests (dry and wet) regardless of the differences in forest structure, soil type, annual precipitation and geographical distance.

$CH_4$ flux data collected long-term in temperate deciduous forest systems in Europe (Butterbach-Bahl and Papen, 2002) has shown that soil moisture can explain up to 58% of the seasonality in soil $CH_4$ uptake. Similarly, Kiese et al. (2003) reported

that soil moisture could explain up to 53% of the seasonality in $CH_4$ exchange in a tropical rainforests in Queensland, Australia. Soil moisture influences soil gas diffusivity and is considered the most important factor controlling seasonality of $CH_4$ uptake in soils worldwide (Dalal et al., 2008;Dalal and Allen, 2008;Smith et al., 2003;Smith et al., 2000;Ball et al., 1997a) and the negative relationship between soil moisture and soil $CH_4$ uptake reported in this study has been previously reported for other ecosystems (Hartmann et al., 2011;Stiehl-Braun et al., 2011;Castro et al., 1994;Price et al., 2003). This

highlights that soil $CH_4$ uptake is mainly diffusion limited in most forest ecosystems (Price et al., 2004) when the sites of microbial $CH_4$ oxidation are distributed through the surface soil, and the concentration gradient, which drives the flux (i.e. atmospheric $CH_4$ concentration), is effectively constant. However, previous field studies have never been able demonstrate so conclusively the strength of the relationship (>90% variation explained) between AFP and soil $CH_4$ uptake, and for two separate forest systems. To our knowledge the only other study where similarly strong correlations between soil moisture

and $CH_4$ uptake have been reported, was for grassland soils under summer rainfall exclusion (Hartmann et al. (2011).

It is important to note that WFPS has commonly been used to model, or compare, soil $CH_4$ uptake in different ecosystems (Del Grosso et al., 2000). However, in our study this soil environmental variable was not as effective as AFP in explaining the observed $CH_4$ flux patterns at the two temperate forest sites. At an individual site level, the relationship between WFPS and $CH_4$ uptake had the same coefficient of determination as between AFP and $CH_4$ uptake, however, the slope of the

relationship differed between the two forest sites (Fig. 4d). This suggests that WFPS is not the most suitable soil moisture metric to relate soil gas diffusivity to soil $CH_4$ flux when comparing sites or ecosystems. This is probably because it is a proportional measure relating VWC to the overall soil porosity; whereas AFP indicates the real volume of air filled pores and therefore diffusion capacity (see equation (4) and (5)). This demonstrates that soil gas diffusivity is primary related to the volumetric fractions of air (AFP), rather than the volumetric fraction of water in the soil since diffusion through air is




much faster than through water. A similar suggestion was made by Farquharson and Baldock (2008) in relation to models of aerobic nitrification processes in soils.

Our data also show a very weak apparent influence by soil temperature upon soil $CH_4$ uptake. However, this temperature effect appears to be mainly driven by the correlation between soil moisture and soil temperature, which is typical for the climate for the investigated forest systems. After the effect of soil moisture was accounted for soil temperature was only able to account for less than 5% of the remaining variability in soil $CH_4$ flux at AU-Wom and less than 1.5% of the remaining variability in soil $CH_4$ flux at AU-Wrr. The coefficient of variance for individual chamber cycles (%$CV_{site}$) at both sites was generally greater than the coefficient of variance for the daily averages (%$CV_{daily}$) which demonstrates that spatial variability in soil $CH_4$ flux within a forest site was greater than daily variability. The daily temperature variation in soil $CH_4$ uptake will have been masked in the analyses because all regression analyses were performed on either chamber cycle or daily uptake means. However, the weak temperature dependency of soil $CH_4$ uptake is unlikely to play a defined role in seasonal variability given that around 90% of seasonal variability in $CH_4$ uptake can be explained by soil moisture alone and that soil moisture and temperature are weakly correlated in the investigated forest systems. This was more pronounced at the AU-Wom site because temporal soil moisture variability was greater and we had two years of data compared to one year of data at the AU-Wrr site.

Our data also clearly show that MOB activity was not limited within the soil moisture range measured during this study. The increase in $CH_4$ uptake was linear for a decrease in WFPS over a 20-60% WFPS range, and linear for an increase in AFP over a 0.3 to 0.53 AFP range. Furthermore, our data also show that soil $CH_4$ uptake still continued at a very low WFPS of 10% (VWC = 0.07 g $cm^{-3}$, AFP = 0.59 $cm^3$ $cm^{-3}$) with $CH_4$ uptake ranging between -62 to – 80 µg $CH_4$ $m^{-2}$ $h^{-1}$ at this time. We can therefore hypothesize that MOB activity was not limited by moisture at the AU-Wom and the AU-Wrr sites during the measurement period.

## 5 Conclusion

Our field data suggest that the difference in magnitude of $CH_4$ flux at both sites was based solely on differences in air filled porosity due to site differences in soil bulk density, soil porosity as the same relationship between AFP and soil $CH_4$ uptake existed at both sites. This means that simple information about soil bulk density could be used to estimate $CH_4$ uptake base rates across different eucalypt forest ecosystems in Australia. Our data further demonstrate that temporal variability in soil $CH_4$ uptake was predominantly controlled by temporal variability in soil AFP that is linked to soil gas diffusivity. This means that seasonality in $CH_4$ uptake can be predicted with very high accuracy where information about soil moisture dynamics is available or can be simulated with high certainty. However, since soil texture at both sites was relatively coarse and soils were both clay loams further studies need to establish if the AFP to $CH_4$ relationship holds true across different soil texture classes. Our results highlight the importance of long-term field measurements in establishing relationships between





soil environmental drivers and soil $CH_4$ uptake and allowing the calibration of models used to calculate global $CH_4$ sink distribution and magnitude.

**Acknowledgements**

The study was supported by funding from the Terrestrial Ecosystem Research Network (TERN) Australian Supersite
5   Network, the TERN OzFlux Network, the Australian Research Council (ARC) grants LE0882936 and DP120101735 and the
Victorian Department of Environment Land, Water and Planning Integrated Forest Ecosystem Research program. We would
like to thank Julio Najera and student volunteers for assistance with site and instrument maintenance.



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



**Table 1: Parameters and coefficients of determination (Adj. R2) of linear regression models explaining seasonal variability in mean chamber cycle methane flux (FCH4) at a mixed Eucalyptus obliqua forest stand, Wombat State Forest, Victoria (AU-Wom) and at a mixed Eucalyptus obliqua and E. regnans forest stand, Warra LTER between, Tasmania, Australia (AU-Wrr). Unstandardised and standardised coefficients β (in parenthesis); SD refers to standard deviation of parameter; level of significance**
5 **as indicated by ANOVA (* ≤ 0.05, ** ≤ 0.01, *** ≤ 0.001). Predictors: TS (Soil temperature), GWC (gravimetric soil water content), AFP (air filled porosity), and VWC (volumetric soil water content)**

| Site | | | | | | |
|---|---|---|---|---|---|---|
| | **Dependent Variable** | *Constant* | *VWC* (SD = 0.051) | $T_S$ (SD = 1.98) | *AFP* (SD = 0.488) | *Adj. $R^2$* |
| AU-Wrr | $F_{CH4}$ (SD = 10.899) | -92.307*** | 195.378*** (0.925) | - | - | 0.855*** |
| | $F_{CH4}$ (SD = 10.899) | -19.543*** | - | -2.215*** (-0.399) | - | 0.158*** |
| | $F_{CH4}$ (SD = 10.899) | -88.835*** | 191.664*** (0.907) | -0.254*** (-0.046) | - | 0.857*** |
| | $F_{CH4}$ (SD = 10.899 | 53.640*** | - | - | -195.378*** (0.925) | 0.855*** |
| | | *Constant* | *VWC* (SD = 0.055) | $T_S$ (SD = 3.42) | *AFP* (SD = 0.402) | *Adj. $R^2$* |
| AU-Wom | $F_{CH4}$ (SD = 11.296) | -75.068*** | 195.768*** (0.957) | - | - | 0.915*** |
| | $F_{CH4}$ (SD = 12.720) | -6.320*** | - | -1.701*** (-0.458) | - | 0.209*** |
| | $F_{CH4}$ (SD = 10.607) | -78.336*** | 201.671*** (0.982) | 0.147*** (0.047) | - | 0.900*** |
| | $F_{CH4}$ (SD = 11.296) | 53.943*** | - | - | 195.768*** (0.957) | 0.915*** |



**Table 2: Parameters and coefficients of determination (Adj. R$^2$) of linear regression models explaining seasonal variability in mean daily methane flux (F$_{CH4}$) at a mixed *Eucalyptus obliqua* forest stand, Wombat State Forest, Victoria (AU-Wom) and at a mixed *Eucalyptus obliqua* and *E. regnans* forest stand, Warra LTER between, Tasmania, Australia (AU-Wrr). Unstandardised and standardised coefficients β *(in parenthesis)*; SD refers to standard deviation of parameter; level of significance as indicated by**
5    **ANOVA ($^*$ ≤ 0.05, $^{**}$ ≤ 0.01, $^{***}$ ≤ 0.001). Predictors: T$_S$ (Soil temperature), GWC (gravimetric soil water content), AFP (air filled porosity), and VWC (volumetric soil water content)**

| Site | | | | | | |
|------|------|------|------|------|------|------|
| | **Dependent Variable** | *Constant* | *VWC* (SD = 0.058) | *T$_S$* (SD = 2.02) | *AFP* (SD = 0.058) | *Adj. R$^2$* |
| AU-Wrr | F$_{CH4}$ (SD = 0.273) | -2.165*** | 4.433*** (0.947) | - | - | 0.896*** |
| | F$_{CH4}$ (SD = 0.273) | -0.459*** | - | -0.052*** (-0.388) | - | 0.148*** |
| | F$_{CH4}$ (SD = 0.273) | -2.167*** | 4.435*** (0.947) | 0.0001 (0.001) | | 0.895*** |
| | F$_{CH4}$ (SD = 0.273) | 1.164*** | - | - | 4.433*** (-0.947) | 0.896*** |
| | | *Constant* | *VWC* (SD = 0.055) | *T$_S$* (SD = 3.55) | *AFP* (SD = 0.055) | *Adj. R$^2$* |
| AU-Wom | F$_{CH4}$ (SD = 0.275) | -1.819*** | 4.771*** (0.962) | - | | 0.924*** |
| | F$_{CH4}$ (SD = 0.302) | -0.161*** | - | -0.038*** (-0.452) | | 0.203*** |
| | F$_{CH4}$ (SD = 0.275) | -1.915*** | 4.956*** (0.999) | 0.004*** (0.053) | | 0.926*** |
| | F$_{CH4}$ (SD = 0.275) | 1.152*** | - | - | -4.771*** (-0.962) | 0.924*** |





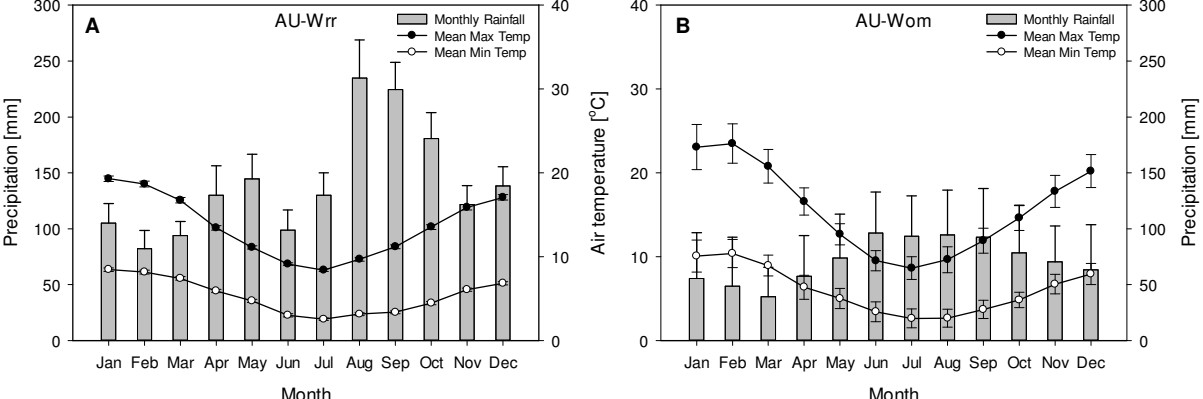

**Figure 1: Climate at the investigated sites: Warra LTER in Tasmania (B, AU-Wrr) and Wombat state Forest in Victoria (A, AU-Wom). Closed symbols represent monthly mean maximum air temperatures, open symbols represent monthly mean minimum air temperatures. Bars represent monthly precipitation. Error bars represent ± 1 SE. Data source Bureau of Meteorology Australia, www.bom.gov.au station numbers 088020 for AU-Wom and 097024 for AU-Wrr.**



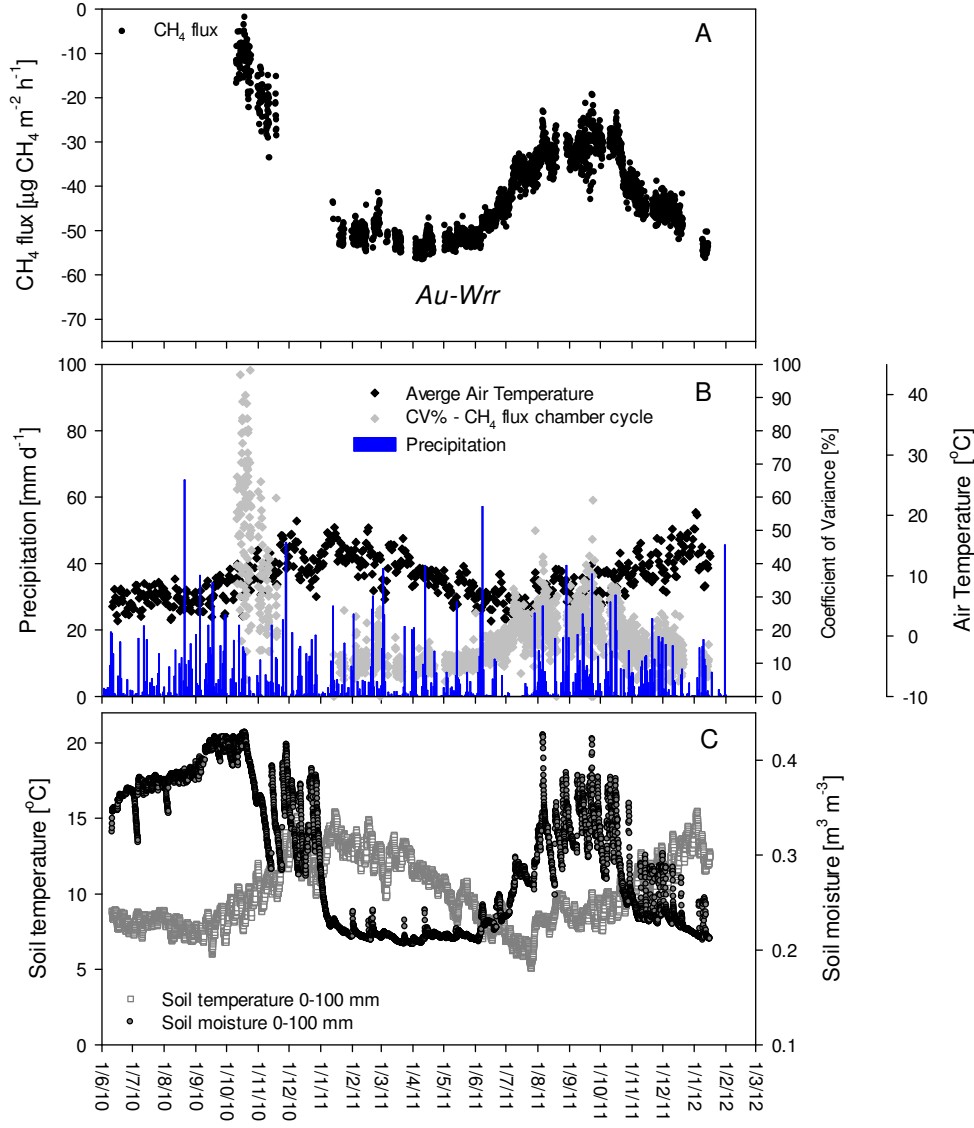

**Figure 2: Soil-based flux of CH$_4$ at a mixed *Eucalyptus obliqua* and *E. regnans* forest stand. Warra LTER, Tasmania, Australia (AU-Wrr). Panel A shows CH$_4$ flux cycle means (10 chambers) per cycle period (4 hours), panel B shows in black closed symbols site air temperature averaged over the chamber cycle period, daily rainfall sums (bars) and coefficient of variance of CH$_4$ flux for each chamber cycle (grey closed symbols). Panel C shows soil temperature in the top 0-10 cm averaged over each chamber cycle (grey open symbols) and corresponding volumetric soil moisture content (grey closed symbols) at the site.**



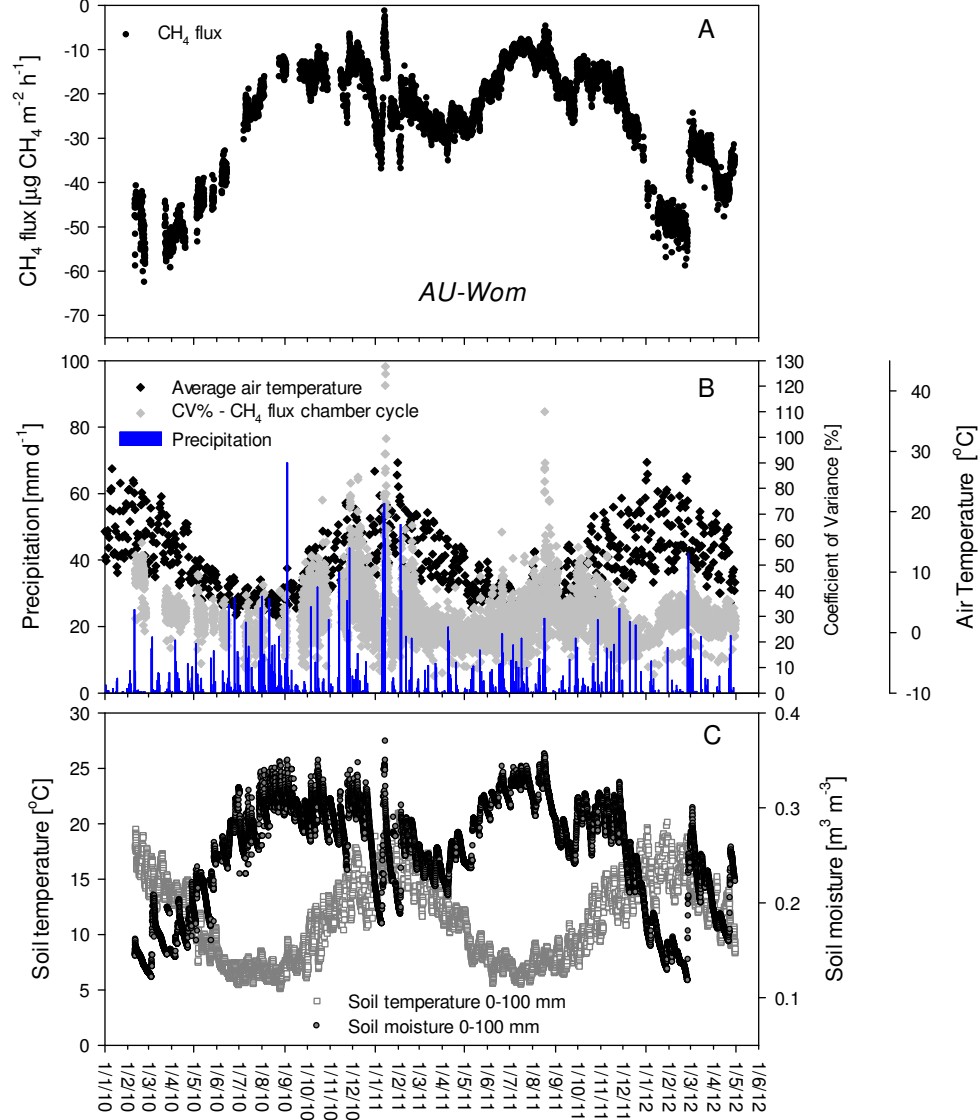

**Figure 3: Soil-based flux of CH$_4$ at a mixed *Eucalyptus obliqua* forest stand, Wombat State Forest, Victoria, Australia (AU-Wom). Panel A shows CH$_4$ flux cycle means (10 chambers) per cycle period (1.5/2 hours), panel B shows in black closed symbols site air temperature averaged over the chamber cycle period, daily rainfall sums (bars) and coefficient of variance of CH$_4$ flux for each chamber cycle (grey closed symbols). Panel C shows soil temperature in the top 0-10 cm averaged over each chamber cycle (grey open symbols) and corresponding volumetric soil moisture content (grey closed symbols) at the site.**





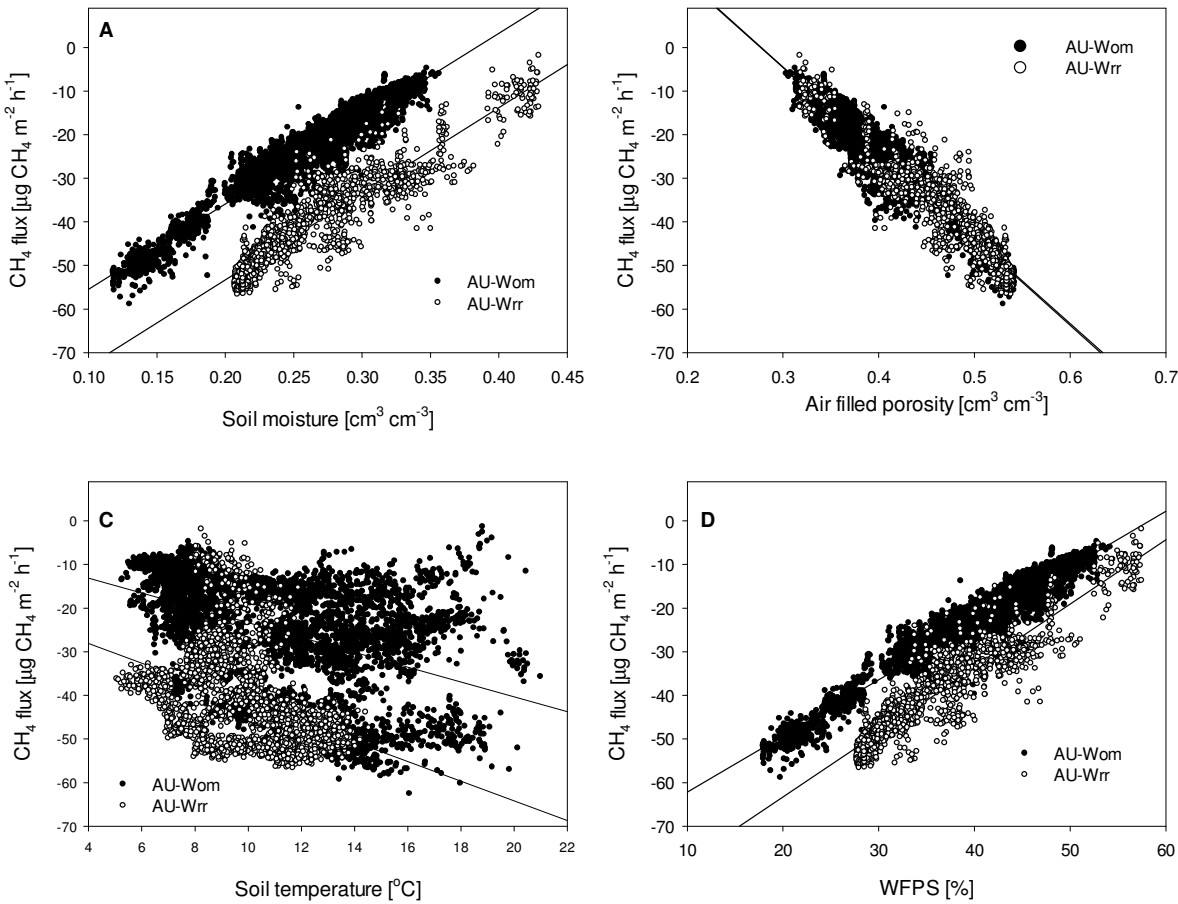

**Figure 4: Relationships between soil volumetric moisture content and soil CH$_4$ flux (A), soil air filled porosity and soil CH$_4$ flux (B), soil temperature and soil CH$_4$ flux (C) and soil water filled pore space (WFPS) and soil CH$_4$ flux for each chamber cycle at a mixed *Eucalyptus obliqua* forest stand, Wombat State Forest, Victoria (closed black symbols, AU-Wom) and at a mixed *Eucalyptus* 5 *obliqua* and *E. regnans* forest stand, Warra LTER between, Tasmania, Australia (open symbols, AU-Wrr). Lines (AU-Wom = solid line; AU_Wrr = dashed line) symbolise significant linear regressions between the parameters (regression parameters are listed in Table 1).**





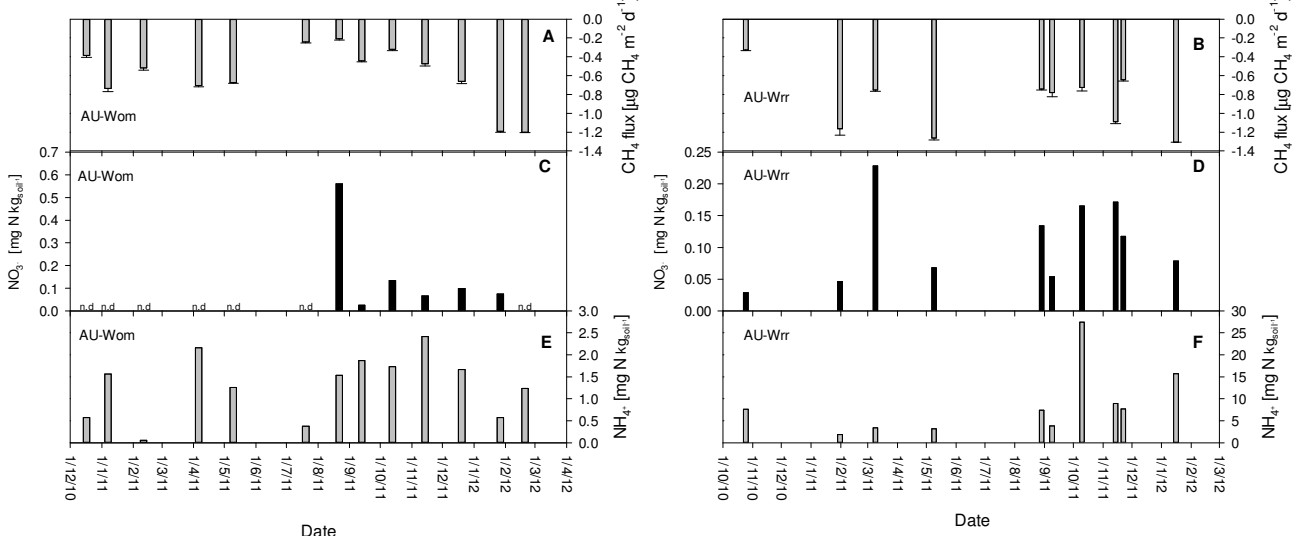

**Figure 5: Dynamics in soil CH$_4$ flux (A, B) soil nitrate levels (C, D) and soil ammonium levels (E, F) at a mixed *Eucalyptus obliqua* forest stand, Wombat State Forest, Victoria (AU-Wom) and a mixed *Eucalyptus obliqua* and *E. regnans* forest stand. Warra**
5 **LTER, Tasmania, Australia. N.d. = not detectable. Not presented are the results of the linear regression analysis between NH$_4^+$ and CH$_4$ flux and NO$_3^-$ and CH$_4$ for both sites, these were: AU-Wom, NO$_3^-$/CH$_4$ (adj. $R^2$ = 0.06, p = 0.21) NH$_4^+$/CH$_4$ (adj. $R^2$ = -0.08, p = 0.83); AU-Wrr NO$_3^-$/CH$_4$ (adj. $R^2$ -0.11, p = 0.80) NH$_4^+$/CH$_4$ (adj. $R^2$ = -0.11, p = 0.84).**