# Peer review of "Soil methane oxidation in both dry and wet temperate eucalypt forests show near identical relationship with soil air-filled porosity"

_Biogeosciences, 2016_

## Referee Comment (RC1) · Anonymous Referee #2 · 30 Jul 2016

Title: Review for bg-2016-181

Fest et al. aimed to understand CH4 dynamics in two Eukaryptus forests in Australia with different precipitation regimes. Fluxes of CH4 were measured in a high temporal resolution with six replications at each site. In addition, soil temperature and moisture, and inorganic N levels were measured. The data were analyzed using linear regression for casual correlation to explore which factors controlled CH4 dynamics. Fest et al. concluded that soil moisture regime could explain over 90% of the variability of CH4 dynamics.

I believe the strengths of this study is 1) very high temporal resolution in CH4 measurements and 2) air-filled porosity explained CH4 dynamics in almost the same manner

for the two study sites. These are novel, and deserve publication. However, the current manuscript is no more than a draft. The weaknesses includes 1) statistical analyses, 2) discussion are underdeveloped, and 3) it's not well written.

1) The data should be analyzed using a maximum likelihood framework with AIC or BIC to compare regressions and determine the importance of temperature. 2) Discussion should emphasize the novelty of this study. 3) I found so many typographical errors throughout the manuscript. Please see the comments below.

Abstract

P1L13-14. Add "under predicted climate change scenarios" to the sentence.

P1L26. Replace "air-filled porosity" with "AFP" as the abbreviation appears in P1L21.

P1L23-25. I disagree with this statement after reading the results and discussion. Activity of MOB was not quantified in this study, and the results cannot indicate MOB activities were similar between the two sites.

P1L24. Check "physiochemical" for difinition. It can be a typo for "physicochemical".

P1L24. Here, the differences between the two sites in $CH_4$ flux were due to "physio-chemical" but AFP explained up to 90% of the variability, indicating that the differences were likely caused by moisture regime.

Introduction

P2L17. Here "air filled" is not hyphenated. Be consistent throughout the manuscript.

P2L33-34. This statement needs citation.

P2L35-P3L1. There are many ecosystems in the Northern Hemisphere without snow or below zero soil temperature, comparable to the Australian forests (e.g., Southwest of USA, Mediterranean region).

Results

P7L23. Replace "around" with "approximately"

P7L23. Fig. 4 should not appear before Fig. 3 (P8L1) in the text.

P7L23. The text mentions 85% (0.85), but Table 1 has 0.896. Why are they different? It's also the case for 19% in the text, and 0.148 in the table.

P7. This is not the best way to analyze the data. Use a model selection approach such as Akaike Information Criterion. For instance, see Monteith et al. (2015):

Monteith DT, Henrys PA, Evans CD, Malcolm I, Shilland EM, Pereira M (2015) Spatial controls on dissolved organic carbon in upland waters inferred from a simple statistical model. Biogeochemistry 123(3): 363-377

P7L30. Avoid starting a sentence using an abbreviation. Spell out AFP.

P8L1. There are AU-Wom and AU-WOM, and AU-Wrr and AU-WRR. Stcik to one form.

P8L20-21. This should be SD, not SE. The large sample size (how many?) makes the SE so small and misleading.

P8L7-9. Awkward sentence, and I cannot find "inter annual" differences were displayed well in figures.

P8L9. VWC in the text, but soil moisture in the figure. Be consistent.

P8L10. Fig. 4a in the text, and Fig. 4A in the figure. Be consistent.

P8L28-P9L7. What is the point of presenting daily $CH_4$ flux in relation to soil environmental variables, if it is not better than that in finer time scales, and does not add much?

P9L5-8. Integrate this section to the first paragraph of the Results.

Discussion P9L10. 1-2 years? I thought the measurements were for two years. Spell out numbers smaller than nine.

Start the discussion on the most exciting findings. I believe the significant correlation between AFP and CH4 flux for the two sites is most interesting in this study. Comparing the daily CH4 flux values with past studies is not too exciting.

P9L25-31. Delete the paragraph. I do not think the statement is true that cool wet temperate eucalypt forests are often compared to rainforests. It's interesting that the annual CH4 flux is comparable between the eucalypt forest and a tropical rainforest, but no more than that. Plus, net CH4 flux is determined by not only MOB activities, but also methanogens as well, especially in wet sites. Thus, there is not much point for the comparison.

P10L1-4. This paragraph should be presented first in the discussion.

P10L10-11. Check the order of the citations.

P10L13-15. Is this an assumption? Delete (i.e. atmospheric CH4 concentration) and add "between soil and atmosphere" after "the concentration gradient" (L14).

P10L21-L23. The coefficient of determination for the relationship between WFPS and CH4 uptake is mentioned in the text, but not shown in Table 1 or 2. The relationship is shown in Fig. 4D, but the coefficient is not shown. If it's discussed in the text, it should be shown somewhere.

P10L28-29. Delete the sentence, and cite Farquharson and Baldock (2008) for the previous sentence.

P10L35-P11L2. This sentence needs to be integrated into the context, otherwise it does not make sense. The paragraph is discussion about temperature on CH4 flux. Then, out of the blue, the sentence on CV of CH4 flux appears without relating it to temperature. It's confusing. I am not quite convinced that temperature did not affect CH4 flux with the current analyses. The better way to test the temperature effect is that 1) construct two models (CH4 flux is a function of moisture, and moisture + temperature) and 2) compare the two models via AIC. This will provide a more concrete

answer.

P11L2. Replace "will" with "would".

P11L9-14. I do not think the statement is valid. First, the authors measured soil moisture only to 10 cm in depth, and did not measure soil moisture in deeper soils. Methanotrophs in deeper soils can contribute to CH4 oxidation if the surface soils are dry. The only way to tease out methanotroph activity from physical constraints of soils for CH4 diffusion is to measure CH4 flux as well as gas diffusivity (see von Fischer et al. 2009).

von Fischer JC, Butters G, Duchateau PC, Thelwell RJ, Siller R (2009) In situ measures of methanotroph activity in upland soils: A reaction‐diffusion model and field observation of water stress. Journal of Geophysical Research: Biogeosciences (2005–2012) 114(G1):

P11L17. Why is "air filled porosity" used here, instead of AFP? Be consistent throughout the manuscript.

P11L17. Replace "same" with "almost identical" (they are not "same" based on Table 1).

P11L18-19. I disagree with the statement. It is possible that AFP governs the CH4 flux across the landscape for eukalypt forests, but there is also a possibility that the casual correlations between AFP and CH4 flux happened to be very similar for the just two study sites. It's not reasonable to extrapolate the results to all the same type of forests in Australia.

Tables

P16L5. "S" in "TS" should be subscript.

Is "-" missing for 195.768 for the AFP parameter at AU-Wom?

"Soil water content" is used in the caption. Is this the same as "soil moisture content" (e.g. P30L6)? If so, use only soil water content.

P17. Table 1. Are "constants" intercepts? Are "parameters" slopes for predictor variables? Are both "unstandardized and standardized coefficients" in parentheses?

Table 1 shows results of four regressions: 1. VWC 2. Soil temp 3. VWC and soil temp 4. AFP

But the corresponding Fig. 4. has: 1. VWC 2. AFP 3. Soil temp 4. WFPS

Why the inconsistency?

Figures

18. I am not sure if the data are presented in the most effective manner in Fig. 1 and 2. The current figures have; A: CH4 flux B: Air temp, CV of CH4 flux, and precip C: Soil temp and soil moisture Is there rationale behind the combinations? I am not convinced that the arrangement makes sense. How about rearrange the combinations; A: CH4 flux and CV B: Precip and soil moisture C. Air and soil temp

Or A: Ch4 flux B: Precip, soil moisture and CV C. Air and soil temp

P18. Fig. 1. AU-Wrr and AU-Wom are Fig. 1A and 1B, respectively, but "A" and "B" letters in the parentheses do not match up. Are these typo?

Replace SE with standard deviations. SE partly depends on the sample size, which is not described in the text, thus the tight error bars can be misleading.

P19. Fig. 2. The description is confusing. Are the individual symbols means of measurements over four hours for each chamber, or average of 10 chambers? Spell out "four" instead of "4".

Replace "moisture" with "water".

In the text (P7 L20), it seems like CV was calculated using average and SD of 10 chambers, but it seems like CV was calculated using average and SD over time for each chamber.

P21. Fig. 4. Add regression equations and Rˆ2 values on the figures.

P22. Fig. 5. I do not think this is the best way to present the data. In the text, the authors want to show there is no significant correlation between CH4 flux and inorganic N contents. Then, scatter plots should be used to show the data.

─────────────────────────────

---

## Referee Comment (RC2) · Anonymous Referee #3 · 29 Aug 2016

The authors investigated the soil methane exchange at two Australian forest study sites differing in annual precipitation. Their major finding is that soil moisture is the main controlling factor and that the relationships of the two sites collapse if air-filled soil porosity, instead of water-filled pore space or volumetric soil moisture is used. The paper is of interest to the readers of BG and the main finding is of general interest to the community as it may trigger new approaches of simulating soil methane exchange if verified across a larger number of sites.

I have three major comments: (1) The study uses two different measurement systems at the two study sites. How can the authors ascertain that the two measurement systems do not cause systematic differences between the two sites? Without a crosscomparison between the two systems at the same site, how can we believe the differences/lack of differences between sites when normalised with AFP? (2) The results of the two sites should be presented together instead of separately for each site. (3) English style and grammar are in the need of checking by a native speaker.

Detailed comments: p. 1, l. 13-14: this sentence comes a bit as a surprise p. 2, l. 2: high compared to many VOC that are present in the ppt range ... p. 2, l. 24: Q10 values critically depend on the depth of the soil temperature used as a reference due to increasing dampening in amplitude and phase shift with soil depth p. 2, l. 28: initiate a new paragraph here p. 2, l. 15-17: would the authors be able to formulate some hypothesis regarding their research? this would strengthen the paper p. 3, l. 28: density p. 7, l. 8: did you check for linearity of tested relationships? p. 7, l. 17: how many longer gaps did you encounter at both sites? p. 7, l. 29-30: figures should be reference in chronological order, i.e. Figure 3 after Figure 2 Results section: the paper would be much more easily readable, if the results of the two sites would be presented together, instead separately – this would help making a stronger point of the major finding of this study; this argument also applies to Figures 2 and 3, which should be combined in my view p. 10, l. 20: diffusion-limited p. 10, l. 22: able to demonstrate p. 10, l. 31-33: reformulate in proper English p. 11, l. 3: I do not get the "However" p. 11, l. 11: what does "defined role" mean?

---

## Author Response (AR1)

**Response to Reviewers – Fest et al., 2016 bg-2016-181**

Reviewer comments appear as normal text

*Our responses appear in bold and italicised*

**Reviewer 1**

Fest et al. aimed to understand CH4 dynamics in two Eucalyptus forests in Australia with different precipitation regimes. Fluxes of CH4 were measured in a high temporal resolution with six replications at each site. In addition, soil temperature and moisture, and inorganic N levels were measured. The data were analyzed using linear regression for casual correlation to explore which factors controlled CH4 dynamics. Fest et al. concluded that soil moisture regime could explain over 90% of the variability of CH4 dynamics.

I believe the strengths of this study is 1) very high temporal resolution in CH4 measurements and 2) air-filled porosity explained CH4 dynamics in almost the same manner for the two study sites. These are novel, and deserve publication.

***We wish to thank the reviewer for a very thorough and constructive review. We agree with most of the suggestions and believe they greatly enhanced the quality of the paper.***

However, the current manuscript is no more than a draft. The weaknesses includes1) statistical analyses, 2) discussion are underdeveloped, and 3) it's not well written.

1) The data should be analyzed using a maximum likelihood framework with AIC or BIC to compare regressions and determine the importance of temperature.

***We agree with the reviewer and have reanalysed the data as suggested by the reviewer: We used a maximum likelihood framework to arrive at the AICs for 3 different models (one model containing only soil temperature, one model containing only a measure of soil moisture (we choose AFP) and one model containing soil temperature and AFP as a predictor. The results of this analysis are displayed in Table R1:***

**Table R1: parameters and coefficients of determination (Adj. R$^2$) of selected linear models in combination with results of a restricted maximum likelihood analysis (REML) explaining seasonal variability in mean chamber cycle methane flux (FCH4) at a mixed *Eucalyptus obliqua* forest stand, Wombat State Forest, Victoria (AU-WOM) and at a mixed E. *obliqua* and *E. regnans* forest stand, Warra LTER between, Tasmania, Australia (AU-WRR). Predictors: T$_S$ (soil temperature) and AFP (air-filled porosity). REML results: Akaike information criterion (AIC); Estimate of importance for models containing both predictors (*in parentheses*).**

| Site | Dependent Variable | *Constant* (Intercept) | *AFP* (slope) | *T$_S$* (slope) | *AIC* | *Adj. R$^2$* |
|------|--------------------|------------------------|---------------|-----------------|-------|--------------|
| AU-WRR | F$_{CH4}$ | 53.640 | -195.378 | - | 5666 | 0.855 |
| | F$_{CH4}$ | -19.543 | - | -2.215 | 9657 | 0.158 |
| | F$_{CH4}$ | 55.587 | -193.284 (***0.997***) | -0.254 (***0.003***) | 5629 | 0.857 |
| AU-WOM | F$_{CH4}$ | 53.943 | -195.768 | - | 7648 | 0.915 |
| | F$_{CH4}$ | -6.320 | - | -1.701 | 13088 | 0.209 |
| | F$_{CH4}$ | 54.766 | -201.671 (***0.998***) | 0.147 (***0.002***) | 7617 | 0.900 |

*The REML and AIC results confirm the interpretation of the original linear regression approach showing that soil moisture (in this case expressed as AFP) is the strongest predictor of soil CH$_4$ flux in both forest systems. The analysis also shows that the models including soil moisture and soil temperature perform marginally better based on AIC compared to models including only soil moisture to predict soil CH$_4$ flux. However, the importance rating of the predictors (soil moisture and soil temperature) clearly indicates that in both forest systems soil moisture dominates accounting more than 99% of the proportion of variance explained by the model compared to <0.01% proportion of the variance explained by soil temperature. This reconfirms our initial assessment of the datasets where we stated that including temperature as a variable improved the correlation with methane uptake to a small degree, but it did not improve the predictive capacity. However, it will be important for readers to understand that and we propose to include the AIC statistics in the revised manuscript.*

*We therefore have added the table above as new Table 2 to the manuscript and added following section to the methods:*

*"We used a restricted maximum likelihood framework (REML, automatic linear modelling in SPSS) to arrive at the Akaike information criterion for three selected models that predict soil $CH_4$ uptake (one model containing only soil temperature, one model containing only a measure of soil moisture (we choose AFP) and one model containing soil temperature and AFP as a predictors of soil $CH_4$ flux)."*

We also added following section to the results:

*"The AIC results of the REML analysis confirm the results of the linear regression approach (Table 2) showing that soil moisture (in this case expressed as AFP) is the strongest predictor of soil $CH_4$ flux in both forest systems. The analysis shows that the models including soil moisture and soil temperature perform marginally better based on AIC compared to models including only soil moisture to predict soil $CH_4$ flux. However, the importance rating of the predictors (soil moisture and soil temperature) clearly indicates that in both forest systems soil moisture dominates, as it accounts for more than 99% of the proportion of variance explained by the model compared to <0.01% proportion of the variance explained by soil temperature."*

2) Discussion should emphasize the novelty of this study.

*We have revised and restructured the discussion to emphasize the novelty of the study:*

*The discussion now reads:*

*"One of the most novel results of our study is the strong linear relationship observed between soil moisture and $CH_4$ uptake. To our knowledge the strength of this relationship is unique for temperate forest systems measured using continuous automated chamber systems over a long-period. It is also striking that this strong linear relationship was similar in the two temperate eucalypt forests (dry and wet) regardless of the differences in forest structure, soil type, annual precipitation and geographical distance. It is possible that the two different measurement systems (GC at AU-WRR and FTIR at AU-WOM) could produce different measures of $CH_4$ flux if operated at the same site because of technological and methodological differences. If that were true, there would only be a remote chance that the two linear relationships between $CH_4$ flux and AFP would overlap one another. As such, our finding that the relationships between $CH_4$ flux and AFP do converge into one common regression line (as shown in Fig. 4) is worthy of note and suggests similar accuracy between the two measurement systems and similar function in soil $CH_4$ exchange processes at the two forest sites.*

[revised manuscript text omitted]

3) I found so many typographical errors throughout the manuscript.

*We apologise for the large number of typographical errors Reviewer 1 discovered and we have endeavoured to correct all to greatly improve the presentation, writing and communication.*

Abstract

P1L13-14. Add "under predicted climate change scenarios" to the sentence.

*This has been added*

P1L26. Replace "air-filled porosity" with "AFP" as the abbreviation appears in P1L21.

*This has been replaced*

P1L23-25. I disagree with this statement after reading the results and discussion. Activity of MOB was not quantified in this study, and the results cannot indicate MOB activities were similar between the two sites.

*We agree and have rephrased this sentence to now read as:*

*"Our data suggest that soil MOB activity in the two forests was similar and that differences in soil CH$_4$ exchange between the two forests were related to differences in soil moisture and thereby soil gas diffusivity."*

P1L24. Check "physiochemical" for definition. It can be a typo for "physicochemical".

*This word has been removed as a consequence of the rephrasing of the sentence above.*

P1L24. Here, the differences between the two sites in CH4 flux were due to "physiochemical" but AFP explained up to 90% of the variability, indicating that the differences were likely caused by moisture regime.

*Soil moisture is a physicochemical difference; it is not a biological difference. However, we have agreed to change the sentence and remove the reference to physicochemical and the sentence now reads as:*

*"Our data suggest that soil MOB activity in the two forests was similar and that differences in soil CH$_4$ exchange between the two forests were related to differences in soil moisture and thereby soil gas diffusivity."*

Introduction

P2L17. Here "air filled" is not hyphenated. Be consistent throughout the manuscript.

*We have standardised this word to appear as "air-filled" throughout.*

P2L33-34. This statement needs citation.

*This statement has citations.*

P2L35-P3L1. There are many ecosystems in the Northern Hemisphere without snow or below zero soil temperature, comparable to the Australian forests (e.g., Southwest of USA, Mediterranean region).

*Reviewer 1 is correct in that there are bioregions in North America that have similar temperature ranges as Australian temperate forest soils. However, this statement is a follow up 'Furthermore,' statement to the primary one, which states that Australian temperate forest soils are highly weathered and very low in nutrients. No change has been made.*

Results

P7L23. Replace "around" with "approximately"

*Agreed, we have replaced.*

P7L23. Fig. 4 should not appear before Fig. 3 (P8L1) in the text.

*Agreed, we have changed the order of the text.*

P7L23. The text mentions 85% (0.85), but Table 1 has 0.896. Why are they different? It's also the case for 19% in the text, and 0.148 in the table.

*Table 1 has 0.855 for VWC and we rounded this to 85%. However the 19% was misquoted and we have corrected this to 16% as represented by and R2 of 0.158 in table 1 for soil temperature.*

P7. This is not the best way to analyze the data. Use a model selection approach such as Akaike Information Criterion. For instance, see Monteith et al. (2015): Monteith DT, Henrys PA, Evans CD, Malcolm I, Shilland EM, Pereira M (2015) Spatial controls on dissolved organic carbon in upland waters inferred from a simple statistical model. Biogeochemistry 123(3): 363-377

*We agree and have re-analysed the data as outlined above. Based on the AIC for each site a model using soil moisture and soil temperature performs marginally better compared to a model using soil moisture alone to predict soil CH₄ uptake (see above). However, the additional amount of CH₄ variance explained by including soil temperature into the model as compared to a model only including soil moisture is less than 1% at both sites. Including temperature in addition to moisture may on a statistical basis improve the model accuracy but not the predictive capacity. Hence, our overall conclusion is still valid. However, we agree that the AIC is the better way of selecting the best model.*

P7L30. Avoid starting a sentence using an abbreviation. Spell out AFP.

*Agreed – we have changed this throughout the manuscript.*

P8L1. There are AU-Wom and AU-WOM, and AU-Wrr and AU-WRR. Stick to one form.

*Agreed, we have now standardised these acronyms to all appear in CAPITALS*

P8L20-21. This should be SD, not SE. The large sample size (how many?) makes the SE so small and misleading.

*We have changed these to Standard Deviations rather than Standard Errors.*

P8L7-9. Awkward sentence, and I cannot find "inter annual" differences were displayed well in figures.

*Agreed, this sentence has been deleted. It effectively replicated, in a confusing way, the statement in the previous sentence and was therefore redundant.*

P8L9. VWC in the text, but soil moisture in the figure. Be consistent.

*We have standardised this to volumetric water content (VWC) throughout the manuscript where appropriate. We use soil moisture as a general term that stands for the different ways soil moisture can be expressed (AFP, VWC and WFPS).*

P8L10. Fig. 4a in the text, and Fig. 4A in the figure. Be consistent.

*We have standardised reference to figures with upper-case e.g. "Fig. 4A"*

P8L28-P9L7. What is the point of presenting daily CH4 flux in relation to soil environmental variables, if it is not better than that in finer time scales, and does not add much?

*We disagree with the reviewer since a lot of available flux studies only ever cite daily flux values and the relationships of daily flux values with environmental variables. We believe that the*

*inclusions of this information in the manuscript will especially be of interest to modellers and will help to put our data in the context of $CH_4$ flux studies globally.*

P9L5-8. Integrate this section to the first paragraph of the Results.

*Given that the annual site $CH_4$ flux budgets are calculated based on the daily flux data presented in the two preceding paragraphs of the result section, we believe it is more logical to leave this section where it is.*

Discussion

P9L10. 1-2 years? I thought the measurements were for two years. Spell out numbers smaller than nine.

*We agree this is confusing. We have changed this to read ">12 months"*

Start the discussion on the most exciting findings. I believe the significant correlation between AFP and CH4 flux for the two sites is most interesting in this study. Comparing the daily CH4 flux values with past studies is not too exciting.

*Agreed. We have rearranged the Discussion sections to open with the discussion of AFP and $CH_4$ flux.*

P9L25-31. Delete the paragraph. I do not think the statement is true that cool wet temperate eucalypt forests are often compared to rainforests. It's interesting that the annual CH4 flux is comparable between the eucalypt forest and a tropical rainforest, but no more than that. Plus, net CH4 flux is determined by not only MOB activities, but also methanogens as well, especially in wet sites. Thus, there is not much point for the comparison.

*We agree with the reviewer and have removed this section form the discussion*

P10L1-4. This paragraph should be presented first in the discussion.

*We agree and now open the Discussion with this paragraph*

P10L10-11. Check the order of the citations.

*We have checked that our citation style is consistent with the BGS style format throughout the manuscript*

P10L13-15. Is this an assumption? Delete (i.e. atmospheric CH4 concentration) and add "between soil and atmosphere" after "the concentration gradient" (L14).

*We have made this suggested change*

P10L21-L23. The coefficient of determination for the relationship between WFPS and CH4 uptake is mentioned in the text, but not shown in Table 1 or 2. The relationship is shown in Fig. 4D, but the coefficient is not shown. If it's discussed in the text, it should be shown somewhere.

*Agreed. We have added coefficients of determination to Figure 4*

P10L28-29. Delete the sentence, and cite Farquharson and Baldock (2008) for the previous sentence.

*Agreed, we deleted the last sentence of the paragraph and placed the citation after the previous sentence.*

P10L35-P11L2. This sentence needs to be integrated into the context, otherwise it does not make sense. The paragraph is discussion about temperature on CH4 flux. Then, out of the blue, the sentence on CV of CH4 flux appears without relating it to temperature. It's confusing. I am not quite convinced that temperature did not affect CH4 flux with the current analyses. The better way to test the temperature effect is that 1) construct two models (CH4 flux is a function of moisture, and moisture + temperature) and 2) compare the two models via AIC. This will provide a more concrete answer.

*In response to the reviewer's suggestion we have removed the sentence about the CV from the paragraph. In response to the second part of the reviewers comment please see our responses above. We have performed the suggested analysis with the result that the AIC indicates that at both sites models including soil temperature and soil moisture perform marginally better as compared to models only including soil moisture to explain $CH_4$ flux variability.*

P11L2. Replace "will" with "would".

*Agreed, this change has been made*

P11L9-14. I do not think the statement is valid. First, the authors measured soil moisture only to 10 cm in depth, and did not measure soil moisture in deeper soils. Methanotrophs in deeper soils can contribute to CH4 oxidation if the surface soils are dry. The only way to tease out methanotroph activity from physical constraints of soils for CH4 diffusion is to measure CH4 flux as well as gas diffusivity (see von Fischer et al. 2009). von Fischer JC, Butters G, Duchateau PC, Thelwell RJ, Siller R (2009) In situ measures of methanotroph activity in upland soils: A reactionˇARˇ diffusion model and field observation of water stress. Journal of Geophysical Research: Biogeosciences (2005–2012) 114(G1):

*We have removed the strong statement that our data clearly demonstrate that there was no moisture limitation of MOB activity at the beginning of this paragraph and now simply make a statement that we didn't see any indication that soil $CH_4$ uptake was moisture limited in our data. This paragraph it now reads:*

*"Furthermore, our data also show that soil $CH_4$ uptake still continued at a very low WFPS of 10% (VWC = 0.07 g cm$^{-3}$, AFP = 0.59 cm$^3$ cm$^{-3}$) with $CH_4$ uptake ranging between -62 to – 80 µg $CH_4$ m$^{-2}$ h$^{-1}$ at this time. We can therefore hypothesize that MOB activity was not severely limited by moisture at the AU-WOM and the AU-WRR sites during the measurement period."*

*However, we also want to point out that the suggested test following the method described by von Fischer 2009 can not necessarily provide the information needed to assess if there is in-situ moisture limitation of methanotrophic activity because this method treats the soil as a one layer and provides a bulk methanotorphic activity measurement and a bulk soil diffusivity measurement without any information on where along the soil profile methanotrophic activity happens at any given point in time. Which means that if as a result of increased diffusivity a larger area of the profile compensates for lower MOB activity in the top soil layer (caused by moisture limitation) it will not show up.*

P11L17. Why is "air filled porosity" used here, instead of AFP? Be consistent throughout the manuscript.

*Agreed, we have changed to AFP*

P11L17. Replace "same" with "almost identical" (they are not "same" based on Table 1).

*Agreed, this change has been made*

P11L18-19. I disagree with the statement. It is possible that AFP governs the CH4 flux across the landscape for eukalypt forests, but there is also a possibility that the casual correlations between AFP and CH4 flux happened to be very similar for the just two study sites. It's not reasonable to extrapolate the results to all the same type of forests in Australia.

*Agreed – this statement was an over-reaching. We have reworded this to now read as:*

*"This means that future research should investigate whether simple information about soil bulk density can be used to estimate $CH_4$ uptake across different eucalypt forest ecosystems in Australia, or in other similar ecosystems globally."*

Tables

P16L5. "S" in "TS" should be subscript.

*Agreed, we have made the subscript change in the Table caption.*

Is "-" missing for 195.768 for the AFP parameter at AU-Wom?

*Yes this has been corrected*

"Soil water content" is used in the caption. Is this the same as "soil moisture content" (e.g. P30L6)? If so, use only soil water content.

*We are using soil moisture content in the manuscript as a general term that represents the three different ways soil moisture content can be expressed (AFP, VWC and WFPS). We believe this is clear and not incorrect.*

P17. Table 1. Are "constants" intercepts? Are "parameters" slopes for predictor variables?

*Yes, to clarify this we have added a sentence to the caption.*

Are both "unstandardized and standardized coefficients" in parentheses?

*We have clarified this in the caption only standardized coefficients are in parentheses*

Table 1 shows results of four regressions: 1. VWC 2. Soil temp 3. VWC and soil temp 4. AFP

But the corresponding Fig. 4. has: 1. VWC 2. AFP 3. Soil temp 4. WFPS Why the inconsistency?

*The reasoning behind displaying WFPS in to $CH_4$ flux relationship in Figure 4 was that we wanted to show that if WFPS is used as a measure of soil moisture the slopes of the relationships with $CH_4$ flux are different at each site. We will add WFPS to the tables 1 and 2*

Figures

18. I am not sure if the data are presented in the most effective manner in Fig. 1 and 2. The current figures have; A: CH4 flux B: Air temp, CV of CH4 flux, and precip C: Soil temp and soil moisture Is there rationale behind the combinations? I am not convinced that the arrangement makes sense. How about rearrange the combinations; A: CH4 flux and CV B: Precip and soil moisture C. Air and soil temp

Or A: Ch4 flux B: Precip, soil moisture and CV C. Air and soil temp

*We arranged the figures this way because the CV% of the $CH_4$ fluxes would convolute figure A to a point where $CH_4$ flux and CV% cannot be separated visually. We decided to leave the figure presentation as it stands.*

P18. Fig. 1. AU-Wrr and AU-Wom are Fig. 1A and 1B, respectively, but "A" and "B" letters in the parentheses do not match up. Are these typo?

*Yes, we have corrected this typographical error in the Figure caption*

Replace SE with standard deviations. SE partly depends on the sample size, which is not described in the text, thus the tight error bars can be misleading.

*Agreed, we have changed to SE to SD in Fig 1A and 1B*

P19. Fig. 2. The description is confusing. Are the individual symbols means of measurements over four hours for each chamber, or average of 10 chambers? Spell out "four" instead of "4".

*Agreed – this can be interpreted several ways. We have rewritten this line in the Figure caption to now read as:*

*"Soil-based flux of $CH_4$ at a mixed Eucalyptus obliqua and E. regnans forest stand. Warra LTER, Tasmania, Australia (AU-WRR). Panel A shows $CH_4$ flux cycle means of ten chambers measured within a four hour time period, panel B shows in black closed symbols site air temperature averaged over the chamber cycle period, daily rainfall sums (bars) and coefficient of variance of the $CH_4$ flux cycle mean shown in Panel A (grey closed symbols). Panel C shows soil temperature in the top 0-10 cm averaged over each chamber cycle (grey open symbols) and corresponding volumetric soil water content (grey closed symbols) at the site*

Replace "moisture" with "water".

In the text (P7 L20), it seems like CV was calculated using average and SD of 10 chambers, but it seems like CV was calculated using average and SD over time for each chamber.

*We agree that this was unclear. CVs are calculated using average and SD of 10 chambers in one measurement period. We have changed the caption to:*

*"Soil-based flux of $CH_4$ at a mixed Eucalyptus obliqua and E. regnans forest stand. Warra LTER, Tasmania, Australia (AU-WRR). Panel A shows $CH_4$ flux cycle means of ten chambers measured within a four hour time period, panel B shows in black closed symbols site air temperature averaged over the chamber cycle period, daily rainfall sums (bars) and coefficient of variance of the $CH_4$ flux cycle mean shown in Panel A (grey closed symbols). Panel C shows soil temperature in the top 0-10 cm averaged over each chamber cycle (grey open symbols) and corresponding volumetric soil water content (grey closed symbols) at the site.*

P21. Fig. 4. Add regression equations and Rˆ2 values on the figures."

*Agreed, the Figure has been annotated with regression equations and R2.*

P22. Fig. 5. I do not think this is the best way to present the data. In the text, the authors want to show there is no significant correlation between CH4 flux and inorganic N contents. Then, scatter plots should be used to show the data.

*We disagree, our intention was to give the reader an idea of the temporal variability in soil nitrate and ammonium concentrations, what can clearly only be achieved with a figure that can accommodate a timeline. The $R^2$ and P values of the regressions between nitrate and ammonium concentrations and $CH_4$ flux are listed in the caption of the figure. We believe that it is quite clear from the figures that there is no significant relationship between these parameters.*

**Reviewer 2**

The authors investigated the soil methane exchange at two Australian forest study sites differing in annual precipitation. Their major finding is that soil moisture is the main controlling factor and that the relationships of the two sites collapse if air-filled soil porosity, instead of water-filled pore space or volumetric soil moisture is used. The paper is of interest to the readers of BG and the main finding is of general interest to the community as it may trigger new approaches of simulating soil methane exchange if verified across a larger number of sites.

I have three major comments:

(1) The study uses two different measurement systems at the two study sites. How can the authors ascertain that the two measurement systems do not cause systematic differences between the two sites? Without a cross- comparison between the two systems at the same site, how can we believe the differences/ lack of differences between sites when normalised with AFP?

*We agree with the reviewer that this is an important point to consider. In our previous research over the last 15 years we have used different measurement systems in many ecosystems in Australia and some of them in the same ecosystem. When using closed-static and closed-dynamic systems in the same ecosystem we never detected large differences in the $CH_4$ flux magnitude. However, we have not been able to test the two automated systems in parallel at the same site. In fact such a comparison has not been conducted anywhere in the literature as far as we are aware. There have been many studies on chamber design and comparisons between automated and manual systems, mainly for $CO_2$ and some for $N_2O$. But a systematic evaluation of an automated closed-static and an automated closed dynamic system for $CH_4$ flux has not been performed. Furthermore there are examples in the literature where data from different manual chamber systems (dynamic and static) and different automated chamber systems (static, dynamic, different analysers) where used a) to compare $CH_4$ flux magnitudes and b) to derive general functional relationships between soil temperature and moisture and soil $CH_4$, fluxes across multiple ecosystems biomes and continents (Curry, 2009, 2007; Dalal and Allen, 2008; Dalal et al., 2008; Del Grosso et al., 2000; Smith et al., 2000). We do not see how our approach is any different. Hence, we acknowledge that it is possible that the two measurement systems could have measured different magnitude of $CH_4$ flux in the two ecosystems and that by chance the relationship between $CH_4$ flux and AFP is identical at the two sites. However, we observed a very strong linear relationship between $CH_4$ flux and AFP at each site and AFP was able to predict around 90% of the flux variation. This is true regardless of the slope of the relationship. So the only difference that the measurement magnitude could make is a different offset of the slope of the relationship. Hence, we include following qualifying statement in the Discussion that highlights this possibility.:*

*"It is possible that the two different measurement systems (GC at AU-WRR and FTIR at AU-WOM) could produce different measures of $CH_4$ flux if operated at the same site because of technological and methodological differences. If that were true, there would only be a remote chance that the two linear relationships between $CH_4$ flux and AFP would overlap one another. As such, our finding that the relationships between $CH_4$ flux and AFP do converge into one common regression line (as shown in Fig. 4) is worthy of note and suggests similar accuracy between the two measurement systems and similar function in soil $CH_4$ exchange processes at the two forest sites."*

*Curry, C. L.: The consumption of atmospheric methane by soil in a simulated future climate, Biogeosciences, 6, 2355-2367, 2009.*

*Curry, C. L.: Modeling the soil consumption of atmospheric methane at the global scale, Global Biogeochemical Cycles, 21, 2007.*

*Dalal, R. C. and Allen, D. E.: Greenhouse gas fluxes from natural ecosystems, Australian Journal of Botany, 56, 369-407, 2008.*
*Dalal, R. C., Allen, D. E., Livesley, S. J., and Richards, G.: Magnitude and biophysical regulators of methane emission and consumption in the Australian agricultural, forest, and submerged landscapes: a review, Plant and Soil, 309, 43-76, 2008.*

*Del Grosso, S. J., Parton, W. J., Mosier, A. R., Ojima, D. S., Potter, C. S., Borken, W., Brumme, R., Butterbach-Bahl, K., Crill, P. M., Dobbie, K., and Smith, K. A.: General $CH_4$ oxidation model and comparisons of $CH_4$ oxidation in natural and managed systems, Global Biogeochemical Cycles, 14, 999-1019, 2000.*

*Smith, K. A., Dobbie, K. E., Ball, B. C., Bakken, L. R., Sitaula, B. K., Hansen, S., Brumme, R., Borken, W., Christensen, S., Prieme, A., Fowler, D., Macdonald, J. A., Skiba, U., Klemedtsson, L., Kasimir-Klemedtsson, A., Degorska, A., and Orlanski, P.: Oxidation of atmospheric methane in Northern European soils, comparison with other ecosystems, and uncertainties in the global terrestrial sink, Global Change Biology, 6, 791-803, 2000.*

(2)  The results of the two sites should be presented together instead of separately for each site.

*We disagree with this suggestion as it is important not to suggest to the readers that these sites can be directly compared. They are independent sites that simply show similar strength of relationship between AFP and $CH_4$ flux. Furthermore, they span difference time periods, experience difference rainfall conditions which would make the figures far more confusing and with less visual resolution.*

(3)  English style and grammar are in the need of checking by a native speaker.

*Agreed. We have extensively edited the manuscript to improve the presentation communication and grammar. Comments from Reviewer 1 greatly assisted in this.*

Detailed comments:

p. 1, l. 13-14: this sentence comes a bit as a surprise

*We have added following sentence ahead of this sentence:*

*"Soils in temperate forest ecosystems are the greatest terrestrial $CH_4$ sink globally."*

p. 2, l. 2: high compared to many VOC that are present in the ppt range ...

*Reviewer 2 is correct. We have changed this sentence to now read as:*

*"Methane (CH₄) has an atmospheric concentration of ~1.8 ppm as compared to >400 ppm for carbon dioxide (CO₂) the second most…….."*

p. 2, l. 24: Q10 values critically depend on the depth of the soil temperature used as a reference due to increasing dampening in amplitude and phase shift with soil depth

***Reviewer 2 is correct. However, the statement still holds as we only build upon and refer to the published literature.***

p. 2, l. 28: initiate a new paragraph here

***Agreed, we have added a paragraph start.***

p. 2, l. 15-17: would the authors be able to formulate some hypothesis regarding their research? this would strengthen the paper

***Our study was objective driven and we do not believe it would be correct to retrospectively add hypothesis to the introduction that would be based on the outcome of the study itself. The dataset is very strong and we firmly believe that the paper as presented it a very valuable contribution this field of research.***

p. 3, l. 28: density

***Yes - we corrected this***

p. 7, l. 8: did you check for linearity of tested relationships?

***Yes, linearity was critical acceptance of chamber flux data. We used an $R^2$ threshold of 0.9 for our quality control.***

p. 7, l. 17: how many longer gaps did you encounter at both sites?

***The figures clearly show when we encountered long data gaps at each site. However, we added a paragraph to the method section outlining the instrument failure related data gap percentage.***

***The paragraph reads:***

***"As outlined above, we excluded fluxes where the coefficient of determination of the regression of chamber concentration versus time was less than 0.9, which lead to the exclusion of approximately 10% of measured chamber fluxes. However, longer gaps in flux data, as shown in Figures 1A and 2A, are either a result of power failures or the need to switch of the power generators on days of extreme fire danger. This led to data gaps of around 30% of the individual datasets."***

p. 7, l. 29-30: figures should be reference in chronological order, i.e. Figure 3 after Figure 2

***Agreed, we have changed the order of the Figures.***

Results section: the paper would be much more easily readable, if the results of the two sites would be presented together, instead separately – this would help making a stronger point of the major finding of this study; this argument also applies to Figures 2 and 3, which should be combined in my view

*We disagree that combining the results of the two sites would greatly assist the reader in understanding the data. We believe that it is important for the reader to clearly see that the relationship between environmental variables and CH₄ flux is identical at each site – not to compare between the two sites. We have trialled this in a previous draft and it was less convincing.*

p. 10, l. 20: diffusion-limited

*We have rephrased this to read as: "limited by diffusion…"*

p. 10, l. 22: able to demonstrate p.

*We rephrased this sentence to:*

*"This agrees with the theory that soil $CH_4$ uptake is mainly limited by diffusion in most forest ecosystems (Price et al., 2004) when the sites of microbial $CH_4$ oxidation are distributed through the surface soil (Stiehl-Braun et al., 2011), and the concentration gradient between soil and atmosphere, which drives the flux , is effectively constant (von Fischer and Hedin, 2007)."*

10, l. 31-33: reformulate in proper English

*This sentence now reads:*

*This is most likely due to the fact that WFPS is a proportional measure that relates VWC to the total soil porosity (equation (4)); compared to AFP that is a direct expression of the air filled pore volume in a given soil (equation (5)).*

p. 11, l. 3: I do not get the "However"

*Agreed. We have deleted "However" at the start of this sentence.*

p. 11, l. 11: what does "defined role" mean?

We have reworded this sentence to now read as:

*"However, the weak temperature dependency of soil $CH_4$ uptake is unlikely to greatly influence seasonal variability given that a…."*

**Soil methane oxidation in both dry and wet temperate eucalypt forests show near identical relationship with soil air-filled porosity**

Benedikt J. Fest[1], Nina Hinko-Najera[2], Tim Wardlaw[3], David W.T. Griffith[4], Stephen J. Livesley[1], Stefan K. Arndt[1]

[1]School of Ecosystem and Forest Sciences, The University of Melbourne, Richmond, 3121 Victoria, Australia
[2]School of Ecosystem and Forest Sciences, The University of Melbourne, Creswick, 3363 Victoria, Australia
[3]Forest Management Services Branch, Forestry Tasmania, Hobart, 7000 Tasmania, Australia
[4]School of Chemistry, University of Wollongong, Wollongong, 2522 New South Wales, Australia

*Correspondence to*: Benedikt J. Fest (bfest@unimelb.edu.au)

**Abstract.** Well-drained, aerated soils are important sinks for atmospheric methane ($CH_4$) via the process of $CH_4$ oxidation by methane oxidising bacteria (MOB). This terrestrial $CH_4$ sink may contribute towards climate change mitigation, but the impact of changing soil moisture and temperature regimes on $CH_4$ uptake is not well understood in all ecosystems. Soils in temperate forest ecosystems are the greatest terrestrial $CH_4$ sink globally. Under predicted climate change scenarios, tTemperate 
[revised manuscript text omitted]
. As outlined above, we excluded fluxes where the coefficient of determination of the regression of chamber concentration versus time was less than 0.9, which lead to the exclusion of approximately 10% of measured chamber fluxes. However, longer gaps in flux data, as shown in Figures 1A and 2A, are either a result of power failures or the need to switch of the power generators on days of extreme fire danger. This led to data gaps of around 30% of the individual datasets.

All statistical analyses were performed with SPSS 20 (IBM, USA). Linear regression procedures and multiple linear regression procedures were used to investigate temporal relationships between measured soil environmental parameters and soil CH$_4$. We initially ran stepwise linear regression procedure as an exploratory tool to identify significant predictors and predictor combinations and retested these afterwards in simple or multiple linear regression models. We transformed data when necessary to reduce heteroscedasticity for linear regression analysis. We used a restricted maximum likelihood framework (REML, automatic linear modelling in SPSS) to arrive at the Akaike information criterion for three selected models that predict soil CH$_4$ uptake (one model containing only soil temperature, one model containing only a measure of soil moisture (we choose AFP) and one model containing soil temperature and AFP as a predictors of soil CH$_4$ flux).

**2.5 Annual site CH$_4$ flux budgets**

To calculate annual site CH$_4$ flux budgets for both sites we first selected a 12 month period with the greatest data coverage for daily average flux for both sites (1/1/2011 – 1/1/2012) and filled existing flux data gaps as follows. For small data gaps of single days where no environmental sensor or flux data were available, we calculated values based on linear interpolation between the CH$_4$ flux of the day before the gap and the day after the gap. For data gaps longer than one day, we used the linear regression model between soil VWC soil moisture and daily soil CH$_4$ flux for each site (Table 1) to estimate the missing CH$_4$ flux data.

**3 Results**

**3.1 CH$_4$ flux in relation to soil environmental variables**

At the AU-Wrr WRR site, soil CH$_4$ flux was always negative indicating CH$_4$ uptake all year around (Fig. 2). The measurement cycle means ranged between -2 µg CH$_4$ m$^{-2}$ h$^{-1}$ (spring 2010) to -58.4 µg CH$_4$ m$^{-2}$ h$^{-1}$ (autumn 2011) with an arithmetic mean of -41.2 ± 0.2311.0 SE SD µg CH$_4$ m$^{-2}$ h$^{-1}$. In general, months with higher average soil moisture and higher

total rainfall displayed lower $CH_4$ uptake when compared to months with lower average soil moisture and lower total rainfall (Fig. 2).  The coefficient of variance (CV) for the average $CH_4$ flux based on 10 chambers in one measurement cycle ranged between 1.8 and 98.0% with an average of 17.9 ± 11% (SD) and was higher in periods of rapid changes in soil moisture levels reflecting changes in precipitation (Fig. 2).

At the AU-WOM site soil $CH_4$ flux was always negative, indicating $CH_4$ uptake all year around (Fig. 3). The measurement cycle means ranged between -1.3 µg $CH_4$ $m^{-2}$ $h^{-1}$ (recorded during a period of heavy rainfall in summer 2011) to -62.5 µg $CH_4$ $m^{-2}$ $h^{-1}$ (summer 2010) with an arithmetic mean of -25.5 ± 12.7 SD µg $CH_4$ $m^{-2}$ $h^{-1}$. Similar to the AU-WRR site, months with higher average soil moisture and higher total rainfall displayed lower $CH_4$ uptake when compared to months with lower average soil moisture and lower total rainfall (Fig. 3). The CV for the average $CH_4$ flux based on six chambers in one measurement cycle ranged between 6.7 and 143.0% with an average of 29.3 ± 9.7% (SD) and was again higher in times of rapid soil moisture changes in response to changes in precipitation patterns (Fig. 3).

For AU-WRR the linear regression analysis showed that volumetric water content (VWC) accounted for  approximately 85% of variability in soil $CH_4$ uptake across all seasons (Fig. 4A, Table 1) with soil $CH_4$ uptake decreasing when soil VWC increased or soil $CH_4$ uptake increasing when air-filled porosity (AFP) increased (Fig. 4B, Table 1). Soil temperature (0-5 cm) alone was weakly related to $CH_4$ uptake with higher $CH_4$ uptake rates associated with higher soil temperatures. However, soil temperature alone was only able to account for approximately 16% of seasonal variability in $CH_4$ uptake (Fig. 4C, Table 1). In addition, after taking the effect of VWC into account, soil temperature only explained around 1.5% of the remaining variability in $CH_4$ uptake at AU-WRR (data not shown). A regression model containing VWC and soil temperature as input variables had only a marginally higher coefficient of determination when compared to the model only containing VWC (Table 1). Air-filled porosity or VWC showed some weak dependency of soil temperature at the site ($R^2 = 0.14$, $p < 0.001$).

~~At the AU WOM site soil $CH_4$ flux was also always negative indicating $CH_4$ uptake all year around (Fig. 3). The measurement cycle means ranged between -1.3 µg $CH_4$ $m^{-2}$ $h^{-1}$ (recorded during a period of heavy rainfall in summer 2011) to -62.5 µg $CH_4$ $m^{-2}$ $h^{-1}$ (summer 2010) with an arithmetic mean of -25.5 ± 0.16 SE µg $CH_4$ $m^{-2}$ $h^{-1}$. Similar to the AU-WRR site months with higher average soil moisture and higher total rainfall displayed lower $CH_4$ uptake when compared to months with lower average soil moisture and lower total rainfall (Fig. 3). The CV for the average $CH_4$ flux based on 6 chambers in one measurement cycle ranged between 6.7 and 143.0% with an average of 29.3 ± 0.12% (SE) and was again higher in times of rapid soil moisture changes in response to changes in precipitation patterns (Fig. 3). Furthermore, inter-annual differences in average site $CH_4$ uptake between seasons were also reflected in concurrently recorded average site soil moisture levels.4a4b~~4B, Table 1). Soil temperature (0-5 cm) alone was again weakly related to $CH_4$

uptake with higher $CH_4$ uptake rates associated with higher soil temperatures (Fig. 4c4C). At the AU-WOM site, only around 20% of seasonal variability in $CH_4$ uptake (Table 1) was explained by soil temperature. In addition, similar to the results at AU-WRRrr, after taking the effect of VWC into account, soil temperature only explained around 5% of the remaining variability in $CH_4$ uptake at AU-Wom WOM (data not shown). Furthermore, a regression model containing VWC and soil temperature had a marginally lower coefficient of determination (Table 1) when compared to the model only containing VWC (Table 1). AFP Air-filled porosity or VWC showed some weak dependency of soil temperature at the site ($R^2$ = 0.38, p < 0.001).

The AIC results of the REML analysis confirm the results of the linear regression approach (Table 2) showing that soil moisture (in this case expressed as AFP) is the strongest predictor of soil $CH_4$ flux in both forest systems. The analysis shows that the models including soil moisture and soil temperature perform marginally better based on AIC compared to models including only soil moisture to predict soil $CH_4$ flux. However, the importance rating of the predictors (soil moisture and soil temperature) clearly indicates that in both forest systems soil moisture dominates, as it accounts for more than 99% of the proportion of variance explained by the model compared to <0.01% proportion of the variance explained by soil temperature.

[revised manuscript text omitted]

Our data also show a very weak influence by soil temperature upon soil $CH_4$ uptake. This temperature effect appears to be mainly driven by the correlation between soil moisture and soil temperature, which is typical for the climate of the investigated forest systems. After the effect of soil moisture was accounted for soil temperature was only able to account for less than 5% of the remaining variability in soil $CH_4$ flux at AU-WOM and less than 1.5% of the remaining variability in soil $CH_4$ flux at AU-WRR. Furthermore, the daily temperature variation in soil $CH_4$ uptake would have been masked in the analyses because all regression analyses were performed on either chamber cycle or daily uptake means. However, the overall weak but statistically significant temperature dependency of soil $CH_4$ uptake is unlikely to greatly influence seasonal $CH_4$ flux variability given that at both sites around 90% of seasonal variability in $CH_4$ uptake can be explained by soil moisture alone and that soil moisture and temperature are weakly correlated in the investigated forest systems. This was more pronounced at the AU-WOM site because temporal soil moisture variability was greater and we had two years of data compared to one year of data at the AU-WRR site. However, a model that includes soil temperature and soil moisture together performed marginally better based on the AIC as compared to a model that only used soil moisture status in predicting soil $CH_4$ flux at both of our sites, which is logical based on the fact that all soil microbial processes show a physiological temperature response but it appears that for the MOB temperature response is rather muted at our sites during our measurement timeframe. 
[revised manuscript text omitted]

| Site | Dependent Variable | Constant (Intercept) | AFP (slope) | $T_S$ (slope) | AIC | Adj. $R^2$ |
|------|--------------------|----------------------|-------------|---------------|-----|-----------|
| AU-WRR | $F_{CH4}$ | 53.640 | -195.378 | - | 5666 | 0.855 |
|  | $F_{CH4}$ | -19.543 | - | -2.215 | 9657 | 0.158 |
|  | $F_{CH4}$ | 55.587 | -193.284 (***0.997***) | -0.254 (***0.003***) | 5629 | 0.857 |
| AU-WOM | $F_{CH4}$ | 53.943 | -195.768 | - | 7648 | 0.915 |
|  | $F_{CH4}$ | -6.320 | - | -1.701 | 13088 | 0.209 |
|  | $F_{CH4}$ | 54.766 | -201.671 (***0.998***) | 0.147 (***0.002***) | 7617 | 0.900 |

Table 3: Parameters and coefficients of determination (Adj. $R^2$) of linear regression models explaining seasonal variability in mean daily methane flux ($F_{CH4}$) at a mixed *Eucalyptus obliqua* forest stand, Wombat State Forest, Victoria (AU-WOM) and at a mixed *Eucalyptus obliqua* and *E. regnans* forest stand, Warra LTER between, Tasmania, Australia (AU-WRR).  Standardised coefficients β *(in parenthesis)*; SD refers to standard deviation of parameter; level of significance  ($^{*} \leq 0.05$, $^{**} \leq 0.01$, $^{***} \leq 0.001$). Predictors: $T_S$ (Soil temperature), , AFP (air-filled porosity), and VWC (volumetric soil water content). Presented constants are model intercepts and parameters represent the slopes for the predictor variables.

| Site | Dependent Variable | Constant | VWC (SD = 0.058) | $T_S$ (SD = 2.02) | AFP (SD = 0.058) | Adj. $R^2$ |
|---|---|---|---|---|---|---|
| AU-W | $F_{CH4}$ (SD = 0.273) | -2.165*** | 4.433*** (0.947) | - | - | 0.896*** |
| | $F_{CH4}$ (SD = 0.273) | -0.459*** | - | -0.052*** (-0.388) | - | 0.148*** |
| | $F_{CH4}$ (SD = 0.273) | -2.167*** | 4.435*** (0.947) | 0.0001 (0.001) | | 0.895*** |
| | $F_{CH4}$ (SD = 0.273) | 1.164*** | - | - | 4.433*** (-0.947) | 0.896*** |
| | | Constant | VWC (SD = 0.055) | $T_S$ (SD = 3.55) | AFP (SD = 0.055) | Adj. $R^2$ |
| AU-WOM | $F_{CH4}$ (SD = 0.275) | -1.819*** | 4.771*** (0.962) | - | | 0.924*** |
| | $F_{CH4}$ (SD = 0.302) | -0.161*** | - | -0.038*** (-0.452) | | 0.203*** |
| | $F_{CH4}$ (SD = 0.275) | -1.915*** | 4.956*** (0.999) | 0.004*** (0.053) | | 0.926*** |
| | $F_{CH4}$ (SD = 0.275) | 1.152*** | - | - | -4.771*** (-0.962) | 0.924*** |

[Figure]

**Figure 1: Climate at the investigated sites: Warra LTER in Tasmania (A, AU-WRR) and Wombat state Forest in Victoria (B, AU-WOM). Closed symbols represent monthly mean maximum air temperatures, open symbols represent monthly mean minimum air temperatures. Bars represent monthly precipitation. Error bars represent ± 1 SD. Data source Bureau of Meteorology Australia, www.bom.gov.au station numbers 088020 for AU-WOM and 097024 for AU-WRR.**

[Figure]

**Figure 2: Soil-based flux of CH$_4$ at a mixed *Eucalyptus obliqua* and *E. regnans* forest stand. Warra LTER, Tasmania, Australia (AU-WRR). Panel A shows CH$_4$ flux cycle means of ten chambers measured within a four hour time period,  panel B shows in black closed symbols site air temperature**

5  **averaged over the chamber cycle period, daily rainfall sums (bars) and coefficient of variance of of the CH$_4$ flux cycle mean shown in Panel A  (grey closed symbols). Panel C shows soil temperature in the top 0-10 cm averaged over each chamber cycle (grey open symbols) and corresponding volumetric soil  water content (grey closed symbols) at the site.**

[Figure]

**Figure 3: Soil-based flux of CH₄ at a mixed *Eucalyptus obliqua* forest stand, Wombat State Forest, Victoria, Australia (AU-WOM).** Panel A shows CH₄ flux cycle means measured within a two hour  time period, panel B shows in black closed symbols site air temperature averaged over the chamber cycle period, daily rainfall sums (bars) and coefficient of variance  CH₄ flux  cycle   (grey closed symbols). Panel C shows soil temperature in the top 0-10 cm averaged over each chamber cycle (grey open symbols) and corresponding volumetric soil water  content (grey closed symbols) at the site.

[Figure]

**Figure 4: Relationships between soil volumetric water content and soil CH$_4$ flux (A), soil air-filled porosity and soil CH$_4$ flux (B), soil temperature and soil CH$_4$ flux (C) and soil water filled pore space (WFPS) and soil CH$_4$ flux for each chamber cycle at a mixed** *Eucalyptus obliqua* **forest stand, Wombat State Forest, Victoria (closed black symbols, AU-WOM) and at a mixed** *Eucalyptus obliqua* **and** *E. regnans* **forest stand, Warra LTER between, Tasmania, Australia (open symbols, AU-WRR). Lines (AU- WOM = solid line; AU_  WRR = dashed line) symbolise significant linear regressions between the parameters (regression parameters are listed in Table 1).**

[Figure]

**Figure 5: Dynamics in soil CH$_4$ flux (A, B) soil nitrate levels (C, D) and soil ammonium levels (E, F) at a mixed *Eucalyptus obliqua* forest stand, Wombat State Forest, Victoria (AU-WOM) and a mixed *Eucalyptus obliqua* and *E. regnans* forest stand. Warra LTER, Tasmania (AU-WRR), Australia. N.d. = not detectable. Not presented are the results of the linear regression analysis between NH$_4^+$ and CH$_4$ flux and NO$_3^-$ and CH$_4$ for both sites, these were: AU-WOM, NO$_3^-$/CH$_4$ (adj. R$^2$ = 0.06, p = 0.21) NH$_4^+$/CH$_4$ (adj. R$^2$ = -0.08, p = 0.83); AU- WRR NO$_3^-$/CH$_4$ (adj. R$^2$ -0.11, p = 0.80) NH$_4^+$/CH$_4$ (adj. R$^2$ = -0.11, p = 0.84).**